# EMPOWERING PROTEIN LANGUAGE MODEL FOR SEQUENCE-STRUCTURE CO-GENERATION WITH CONTINUOUS STRUCTURE TOKENS

## ABSTRACT

Proteins inherently possess a consistent sequence-structure duality. The abundance of protein sequence data, which can be readily represented as discrete tokens, has enabled fruitful developments in protein language models (pLMs). A key remaining challenge, however, is how to effectively integrate continuous structural knowledge into pLMs. Current methods often discretize protein structure to accommodate the language modeling framework, which inevitably results in fine-grained information loss and limits the performance potential of multi-modal pLMs. In this paper, we argue that such concerns can be circumvented: a sequence-based pLM can be extended to incorporate the structure modality through continuous tokens, i.e., high-fidelity protein structure latents that avoid vector quantization. Specifically, we propose a hybrid diffusion protein language model, **HD-Prot**, which embeds a continuous-valued diffusion generation head atop a discrete pLM, enabling seamless operation with both discrete and continuous tokens for sequence-structure joint-modeling in multimodal generative pLMs. The proposed model captures inter-token dependencies across modalities through a unified absorbing diffusion process, and estimates per-token distributions via categorical prediction for sequences and continuous diffusion for structures. Extensive empirical results show that our models achieve competitive performance in unconditional sequence–structure co-generation, motif-scaffolding, protein structure prediction, and inverse folding tasks, performing on par with state-of-the-art multimodal pLMs despite being developed under limited computational resources. It underscores the viability of jointly modeling discrete categorical and continuous arbitrary distributions using shared parameters within a pLM, pointing to an alternative and promising direction of progress for multimodal pLMs.

## 1 INTRODUCTION

Proteins, as the fundamental workhorses of life, orchestrate nearly all cellular processes. Their biological roles are governed by a canonical paradigm (Anfinsen, 1973; Fan et al., 2025) – the **amino acid sequence** of a protein determines its **three-dimensional structure**, which in turn defines its function. As illustrated on the left of Figure 1, this relationship highlights both the intrinsic synergy and the distinct nature of protein sequences and structures. Although they are strongly correlated in a biological sense, they exhibit significant divergence in data modality: the sequence comprises a **discrete** arrangement of amino acid types, whereas the structure is described by **continuous**-valued 3D coordinates. This drives an ambitious goal: to build up a unified protein generative model that comprehensively estimates the joint distribution of protein se-

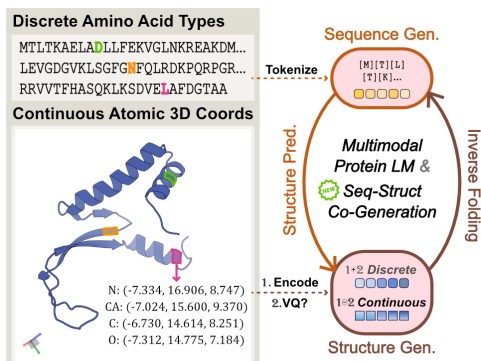

Figure 1: Background and Motivation. Left: An example of protein sequence and structure. Right: Multimodal pLMs enable joint sequence-structure learning, yet face a fundamental choice in protein structure representation.

quences and structures. Benefiting from the greater scale of protein sequence data and the remarkable success of language model pre-training, protein language models (**pLMs**) have established a robust foundation for exploring the vast protein universe. Subsequently, an effective path towards the joint learning of protein sequence and structure is to perform modal extension on pLMs (Hayes et al., 2025; Wang et al., 2024b). These models leverage their strong sequence knowledge, facilitating the consistent structure learning through the shared parameters and a unified semantic space. Therefore, as shown on the right of Figure 1, multimodal pLMs have the capability to complete complex cross-modal tasks, especially the sequence-structure co-generation as an all-in-one protein design solution.

Nevertheless, as multimodal generative pLMs continue to advance, a critical design choice remains in how to represent protein structure. To align with standard language model architectures, existing prominent approaches often opt to process the protein structure into discrete tokens. Concretely, ESM3 (Hayes et al., 2025) and DPLM-2 (Wang et al., 2024b) introduce protein structure tokenizers based on quantizers like VQ-VAE (Van Den Oord et al., 2017; Yu et al., 2023), thereby representing each structure as a sequence of discrete tokens from learned codebooks. However, a fundamental limitation remains here: the quantization process inevitably compresses and omits portions of continuous information, leading to the loss of fine-grained structural details and imprecise geometric relationships. This information loss impairs the reconstruction capability of the structural tokenizer first, and ultimately caps the achievable accuracy of structure modeling in multimodal protein language models. Having noticed this issue, DPLM-2.1 (Hsieh et al., 2025) further increases the granularity of discrete structure tokens through bit-wise quantization and diffusion-based residual recovery. However, the information loss caused by protein structure discretization still exists, and the combination of quantization and recovery modules also brings engineering inconvenience.

As a promising alternative to discretization approaches, there has been a recent trend toward embracing *continuous tokens* in many multimodal domains, particularly visual-language modeling (Wang et al., 2024a; 2025b), with the aim of enhancing continuous information fidelity. For example, Chen et al. (2025) presents an efficient continuous image tokenizer that achieves a high compression ratio while enhancing the semantic richness of the latent space. Li et al. (2024) suggests that auto-regressive modeling does not necessarily need to be coupled with discrete and vector-quantized representations. High-quality image generation can be achieved through autoregressive modeling of per-token probability distributions in a continuous-valued space. Furthermore, Fan et al. (2024) reveals that quantization-based models exhibit slower performance improvements in visual tasks when scaling up model size, compared to models operating on continuous tokens. In a nutshell, the effectiveness of utilizing continuous tokens has been demonstrated in the visual-language modeling tasks, benefiting from their expressive capability in representing fine-grained knowledge. Inspired by these cutting-edge advancements, embracing continuous structure tokens alongside natively discrete sequence tokens holds promise for empowering pLMs to achieve high-quality modeling of both protein sequences and structures. In this context, a research question arises: *Can a protein language model capture structural information in a continuous space, while preserving the extensive discrete sequence knowledge?*

In this study, we conclusively address this question with an affirmative answer. To be specific, we propose **HD-Prot**, a **h**ybrid **d**iffusion framework that extends a sequence-only protein language model into a multimodal pLM by incorporating continuous structure tokens. Figure 2 presents the overall architecture of the proposed HD-Prot. First, a non-quantized autoencoder is introduced as the protein structure tokenizer, where latent embeddings that can be highly accurately reconstructed into structural coordinates are considered as continuous structure tokens. At a high level, HD-Prot places the continuous structure tokens on an equal footing with the discrete sequence tokens. Diffusion language modeling is applied in parallel to both token tracks, involving a noising process that masks sequence and structure tokens, followed by a generation process of iterative mask token prediction. More concretely, the sequence-structure information is residue-wise integrated from the very beginning and consistently processed by the main body of a protein language model. Then, the per-token probability distribution is estimated via language modeling in categorical space for sequence and diffusion modeling in continuous space for structure.

In summary, our main contributions are highlighted as follows:

- This paper highlights the promising potential of using continuous tokens to represent protein structure information within protein language models (pLMs). We demonstrate that it is effective and

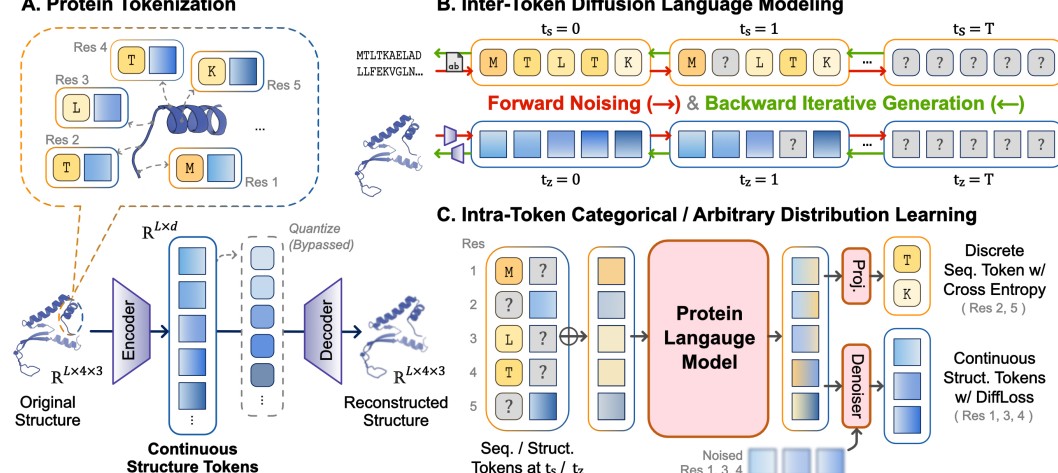

Figure 2: Overview of the proposed HD-Prot. (**A**) Protein backbone structure is processed into continuous tokens via an advanced non-quantized tokenizer. Each protein is represented by a track of discrete sequence tokens and a track of continuous structure tokens, aligned residue-wise. (**B**) HD-Prot performs diffusion language modeling to capture inter-token dependencies, wherein both sequence tokens and structure tokens are noised by and denoised from the absorbing state, i.e., the mask. (**C**) HD-Prot models the protein sequence and structure almost within a unified pLM. Based on the hidden states produced by the pLM, we introduce a categorical head for discrete sequence modeling and a denoising head for continuous-valued structure generation.

efficient to develop a multimodal generative pLM with a non-quantized structure tokenizer and a publicly available sequence-only pre-trained pLM.

- We propose HD-Prot, a hybrid diffusion framework that bridges the discrete-continuous modality gap in multimodal protein modeling. In addition to the unified absorbing diffusion language modeling at the inter-token level, the key lies in differentiating the learning of sequence and structure knowledge at the intra-token level. Alongside the categorical mask prediction performed on discrete sequence tokens, HD-Prot estimates the probability distribution of continuous structure tokens via a diffusion procedure operating on a continuous-valued domain.

- We present extensive quantitative and qualitative experiments on unconditional sequence-structure co-generation, motif-scaffolding, protein structure prediction, and inverse folding tasks. HD-Prot models show strong competitiveness compared to representative multimodal pLMs, exhibiting a notable ability to estimate the joint distribution of protein sequence and structure. Furthermore, our study reveals several valuable insights into practical implementation, specifically regarding robust modality expansion, classifier-free guidance for continuous structure tokens, and efficient low-cost training.

## 2 PRELIMINARIES

**Multimodal Protein Modeling.** A protein can be comprehensively characterized through its sequence and structure. For a protein with $L$ residues, its sequence is defined as $\boldsymbol{s} = (s_1, s_2, \ldots, s_L)$, where each $s_i$ $(1 \leq i \leq L)$ is a categorical variable denotes the amino acid identity of the $i$-th residue, generally involved in 20 standard amino acids $\mathbb{S}^{20} = \{\texttt{A}, \texttt{R}, \ldots, \texttt{V}\}$. Meanwhile, the protein structure is represented as $\boldsymbol{x} = (x_1, x_2, \ldots, x_L)$, where $x_i \in \mathbb{R}^{n_i \times 3}$ encoding the Cartesian coordinates of all atoms in the $i$-th residue. We specifically consider backbone atoms $\{\text{N}, \text{C}_\alpha, \text{C}, \text{O}\}$ that captures the essential structural scaffold, thus simplifying each $x_i$ to a real-value matrix in $\mathbb{R}^{4 \times 3}$.

Generative modeling estimates the probabilistic distribution of protein data via a neural network $\theta$. It's expected that a multimodal protein model can holistically understand and explore the protein universe, estimating the joint sequence-structure distribution natively, expressed formally as:

$$p_\theta(\text{Protein}) = p_\theta(\boldsymbol{s}, \boldsymbol{x}) = p_\theta(s_1, s_2, \ldots, s_L, \ x_1, x_2, \ldots, x_L). \quad (1)$$

Whereupon, we are able to perform protein sequence-structure co-generation straightforwardly, and conduct conditional generations across modalities (Wang et al., 2025a; Campbell et al., 2024; Wang et al., 2024b; Meshchaninov et al., 2024). Such an all-in-one modeling framework is opening up a new direction beyond the cascaded calls of independent sequence/structure generation (Wang et al., 2024a; Watson et al., 2023; Geffner et al., 2025b; Lin et al., 2024), structure prediction (Jumper et al., 2021; Lin et al., 2023), and inverse-folding (Dauparas et al., 2022; Hsu et al., 2022) models.

**Diffusion Language Models.** Diffusion models (Ho et al., 2020; Karras et al., 2022; Song et al., 2020) learn to synthesize data by gradually denoising random noise through an iterative process that reverses a predefined noise-adding Markov chain. Significant breakthroughs first emerged in the image domain, where diffusion models learn to estimate arbitrary continuous-valued data distributions through iterative denoising of Gaussian noise. Recent advances have extended diffusion models to language modeling (DeepMind, 2025; Nie et al., 2025; Yu et al., 2025), achieving strong performance across a range of benchmarks. When adopting the mask token `<Mask>` as an *absorbing* state, diffusion language models operating on categorical distributions retain the basic idea of diffusion models. Here we illustrate it following the formulations of Wang et al. (2024a).

The ***forward process*** progressively corrupts an input sentence $\boldsymbol{s}^{(0)}$ over $T$ diffusion steps through iterative token masking, ultimately transforming all tokens into the mask token. The $t$-step marginal distribution admits:

$$q(\boldsymbol{s}^{(t)}|\boldsymbol{s}^{(0)}) = \text{Cat}\left(\boldsymbol{s}^{(t)}; \bar{\alpha}_t \boldsymbol{s}^{(0)} + (1 - \bar{\alpha}_t)\boldsymbol{q}_{\text{noise}}\right), \tag{2}$$

where $\boldsymbol{q}_{\text{noise}}$ is a fixed probability vector concentrated on the mask token, and $\bar{\alpha}_t$ represents the preservation rate of original tokens determined by a masking schedule, satisfying $\bar{\alpha}_t \to 0$ as $t \to T$. The ***reverse process*** is learned by parameterizing the denoising transition steps:

$$p_\theta\left(\boldsymbol{s}^{(t-1)}|\boldsymbol{s}^{(t)}\right) = \sum_{\hat{\boldsymbol{s}}^{(0)}} q\left(\boldsymbol{s}^{(t-1)}|\boldsymbol{s}^{(t)}, \hat{\boldsymbol{s}}^{(0)}\right) p_\theta(\hat{\boldsymbol{s}}^{(0)}|\boldsymbol{s}^{(t)}), \tag{3}$$

where $\hat{\boldsymbol{s}}^{(0)}$ denotes the model's prediction of the full sentence, and transition kernel $q\left(\boldsymbol{s}^{(t-1)}|\boldsymbol{s}^{(t)}, \hat{\boldsymbol{s}}^{(0)}\right)$ samples a less noisy $\boldsymbol{s}^{(t-1)}$ based on the $\boldsymbol{s}^{(t)}$ and $\hat{\boldsymbol{s}}^{(0)}$. As simplified by Austin et al. (2021), the ***training*** undergoes a reweighted masked language modeling:

$$\mathcal{L} = \mathbb{E}_{\boldsymbol{s}^{(0)}}\left[\lambda^{(t)} \sum_{i=1}^{L} \mathbf{1}_{\boldsymbol{s}_i^{(t)} = \texttt{<Mask>}} \log p_\theta(\boldsymbol{s}_i^{(0)}|\boldsymbol{s}^{(t)})\right], \tag{4}$$

where $L$ represents the length of the corpus and $\lambda^{(t)}$ is a reweighting term induced from specific masking schedules. Eventually, ***generation*** begins with a sequence of `<Mask>` of a specified length, and progressively approaches the realistic sequence $\boldsymbol{s}^{(0)}$ by iterative mask token prediction and remasking that selectively adopts a subset of predicted tokens at each step.

## 3 THE PROPOSED METHOD: HD-PROT

Figure 2 provides an overview of the proposed HD-Prot. Firstly, there is a protein structure tokenizer capable of transforming between the 3D coordinates and the latent embeddings, i.e., continuous protein structure tokens. Subsequently, our HD-Prot framework extends a sequence-only protein language model into a multimodal model by integrating the additional continuous structure tokens.

### 3.1 CONTINUOUS PROTEIN STRUCTURE TOKENS

As shown in Figure 2.A, a 3D protein structure and its continuous structure tokens are interconverted by a tokenizer. It basically operates like a non-quantized protein autoencoder, following the encoding-decoding process: $\boldsymbol{x} \xrightarrow{\text{encoder}} \boldsymbol{z} \xrightarrow{\text{decoder}} \hat{\boldsymbol{x}}$, where $\boldsymbol{x} \in \mathbb{R}^{L \times 4 \times 3}$ is the input backbone structure, $\boldsymbol{z} \in \mathbb{R}^{L \times d_{\text{struct}}}$, $d_{\text{struct}} = 20$ is the continuous tokens, and $\hat{\boldsymbol{x}} \in \mathbb{R}^{L \times 4 \times 3}$ is the reconstructed structural coordinates. Moreover, as the foundation for subsequent lan-

Table 1: Structure Recon. Quality

| Tokenizer | $|\mathcal{Z}|$ | CAMEO | |
| | | scRMSD $\downarrow$ | scTM $\uparrow$ |
| --- | --- | --- | --- |
| DPLM-2 | 8092 | 2.109 ± 2.635 | 0.940 ± 0.080 |
| ESM3 | 4096 | 1.225 ± 2.485 | 0.975 ± 0.072 |
| salad-vq | 4096 | 0.842 ± 0.913 | 0.981 ± 0.042 |
| **salad** | - | 0.393 ± 1.072 | 0.996 ± 0.015 |

guage modeling, an ideal structure tokenizer should learn the structural *equivalence* and *contextual locality* for proteins. Specifically, equivalence ensures the structure tokens $\boldsymbol{z}$ are invariant to the global rotation/translation of $\boldsymbol{x}$, enabling the use of a standard, non-equivariant transformer for pLMs. Contextual locality means each $\boldsymbol{z}_i$ (for $1 \leq i \leq L$) primarily corresponds to the local structural environment of residue $i$, ensuring that masking it forces the pLM to learn effective context rather than exploiting global shortcuts. To satisfy these requirements, an advanced protein structure autoencoder named salad (Jendrusch & Korbel, 2025), featuring a sparse invariant point attention (IPA) architecture, is introduced as our protein structure tokenizer.

The primary motivation for introducing continuous structure tokens is to minimize information loss in protein structure representation, which could be validated by high-quality protein structure reconstruction. As shown in Table 1, the salad tokenizer outperforms DPLM-2 (Wang et al., 2024b) and ESM3 (Hayes et al., 2025) tokenizers on the CAMEO 2022 test set, while also significantly surpassing its VQ-version counterpart (Jendrusch & Korbel, 2025), demonstrating the advantage of avoiding quantization. The near-perfect reconstruction capability of the salad tokenizer demonstrates that its resulting continuous tokens retain virtually all essential structural information. See Appendix B for more detailed analysis.

## 3.2 HYBRID DIFFUSION PROTEIN LANGUAGE MODEL

We propose a hybrid diffusion framework for multimodal protein modeling, which enables a pLM to jointly model a track of discrete sequence tokens $\boldsymbol{s} = (s_1, s_2, \ldots, s_L)$ and a track of continuous structure tokens $\boldsymbol{z} = (z_1, z_2, \ldots, z_L)$. The common per-residue tokenization allows for a unified absorbing diffusion language modeling at the inter-token level, while the discrete/continuous distinction requires separate estimation of categorical/arbitrary distributions at the intra-token level.

**Inter-Token Diffusion Language Modeling.** Protein sequences and structures embody a wealth of evolutionary, functional, and folding knowledge, reflected in the relationships between sequence tokens, between structure tokens, and across modalities. HD-Prot perform unified diffusion language modeling to learn this rich protein knowledge, simultaneously capturing bidirectional contextual dependencies within each modality and cross-modal alignments.

Fundamentally, HD-Prot introduces absorbing states via dedicated mask tokens: $\boldsymbol{m}_s$ for sequences and $\boldsymbol{m}_z$ structures. It configures decoupled schedulers $t_s \in \{0, 1, \ldots, T\}$ and $t_z \in \{0, 1, \ldots, T\}$ for the sequence and structure, respectively. Distinct configurations of the two schedulers drive diverse protein modeling tasks, which are detailed in the Appendix C.1. As illustrated in Figure 2.B, the *forward process* graudally noise the initial sequence and structure tokens $\left(\boldsymbol{s}^{(0)}, \boldsymbol{z}^{(0)}\right)$ into masks via limited diffusion steps. States at the combined $(t_s, t_z)$ step is formally defined as:

$$q(\boldsymbol{s}^{(t_s)}|\boldsymbol{s}^{(0)}) = (1 - \frac{t_s}{T})\boldsymbol{s}^{(0)} + \frac{t_s}{T}\boldsymbol{m}_s, \quad q(\boldsymbol{z}^{(t_z)}|\boldsymbol{z}^{(0)}) = (1 - \frac{t_z}{T})\boldsymbol{z}^{(0)} + \frac{t_z}{T}\boldsymbol{m}_z. \tag{5}$$

For the sequence track, $(\frac{t_s}{T})L$ randomly selected tokens are replaced with mask token $\boldsymbol{m}_s$ and the remaining $(1 - \frac{t_s}{T})L$ tokens are preserved from the original $\boldsymbol{s}^{(0)}$; so as for the structure track. Given that, the model learns to denoise from the fully masked state $\left(\boldsymbol{s}^T, \boldsymbol{z}^T\right)$ through a parameterized *reverse process*:

$$p_\theta\left(\boldsymbol{s}^{(t_s-1)}|\boldsymbol{s}^{(t_s)}, \boldsymbol{z}^{(t_z)}\right) = \sum_{\hat{\boldsymbol{s}}^{(0)}} q\left(\boldsymbol{s}^{(t_s-1)}|\boldsymbol{s}^{(t_s)}, \hat{\boldsymbol{s}}^{(0)}\right) p_\theta(\hat{\boldsymbol{s}}^{(0)}|\boldsymbol{s}^{(t_s)}, \boldsymbol{z}^{(t_z)}),$$

$$p_\theta\left(\boldsymbol{z}^{(t_z-1)}|\boldsymbol{z}^{(t_z)}, \boldsymbol{s}^{(t_s)}\right) = \sum_{\hat{\boldsymbol{z}}^{(0)}} q\left(\boldsymbol{z}^{(t_z-1)}|\boldsymbol{z}^{(t_z)}, \hat{\boldsymbol{z}}^{(0)}\right) p_\theta(\hat{\boldsymbol{z}}^{(0)}|\boldsymbol{z}^{(t_z)}, \boldsymbol{s}^{(t_s)}). \tag{6}$$

For the sequence track, $\hat{\boldsymbol{s}}^{(0)}$ denotes the model's prediction of the initial state based on the partially masked states at $(t_s, t_z)$, and the less noisy $\boldsymbol{s}^{(t_s-1)}$ is sampled conditioned on the $\left(\boldsymbol{s}^{(t_s)}, \boldsymbol{z}^{(t_z)}\right)$ and $\hat{\boldsymbol{s}}^{(0)}$ via the transition kernel $q$; so as for the structure track.

**Intra-Token Categorical / Arbitrary Distribution Learning.** To accommodate the distinct characteristics of the multimodal data, we introduce two intra-token learning channels: categorical prediction for sequence tokens and continuous-valued estimation for structure tokens. As shown in Figure 2.C, the partially masked sequence and structure tokens are fused at the input and processed through a protein language model (pLM):

$$\boldsymbol{c} = \text{pLM}\left(\boldsymbol{c}_{\text{seq}} + \boldsymbol{c}_{\text{struct}}\right), \quad \boldsymbol{c}_{\text{seq}} = \text{embed}\left(\boldsymbol{s}^{(t_s)}\right), \quad \boldsymbol{c}_{\text{struct}} = \text{norm}\left(\boldsymbol{z}^{(t_s)}\right) W_{\text{in}}, \tag{7}$$

where the sequence tokens $\boldsymbol{s}^{(t_s)}$ are mapped to embeddings $\boldsymbol{c}_{\text{seq}} \in \mathbb{R}^{L \times d_{\text{hidden}}}$ via the pLM's embedding module, and the structure tokens $\boldsymbol{s}^{(t_z)}$ are transformed to $\boldsymbol{c} \in \mathbb{R}^{L \times d_{\text{hidden}}}$ via a layer normalization and a linear projection $W_{\text{in}} \in \mathbb{R}^{d_{\text{struct}} \times d_{\text{hidden}}}$. A pLM receives the fused sequence-structure representation and produces the deeply integrated protein representation $\boldsymbol{c}$. Together, the element-wise summation operation and the shared language model position encoding guarantee *residue-by-residue sequence-structure alignment* (Hayes et al., 2025).

Subsequently, the model learns to estimate the *per-token distribution* through a reweighted cross-entropy loss and diffusion loss (Li et al., 2024) for sequence and structure tokens, repsectively:

$$\mathcal{L}_{\text{seq}} = \mathbb{E}_{\boldsymbol{s}^{(0)}} \left[ \lambda_{\text{seq}}^{(t_s)} \sum_{i=1}^{L} \mathbf{1}_{\boldsymbol{s}_i^{(t_s)} = \boldsymbol{m}_s} \log p(\boldsymbol{s}_i^{(0)} | \boldsymbol{c}_i) \right], \ p(\boldsymbol{s}_i^{(0)} | \boldsymbol{c}_i) = \text{Softmax} \left( \text{Projector} \left( \boldsymbol{c}_i \right) \right); \quad (8)$$

$$\mathcal{L}_{\text{struct}} = \mathbb{E}_{\boldsymbol{z}^{(0)}} \left[ \lambda_{\text{struct}}^{(t_z)} \sum_{i=1}^{L} \mathbf{1}_{\boldsymbol{z}_i^{(t_z)} = \boldsymbol{m}_z} \| \epsilon - \hat{\epsilon}_i \|^2 \right], \ \hat{\epsilon}_i = \text{Denoiser} \left( \sqrt{\bar{\alpha}_{t'}} \boldsymbol{z}_i^{(0)} + \sqrt{1 - \bar{\alpha}_{t'}} \epsilon, t', \boldsymbol{c}_i \right); \quad (9)$$

where the Projector predicts the categorical logits over the vocabulary of sequence tokens, while the Denoiser is a noise predictor under the typical DDPM framework (Ho et al., 2020). $\lambda^{(t_s)}$ and $\lambda^{(t_z)}$ are reweighting coefficients that control the trade-off between micro and macro perceptions during sequence and structure modeling. For residue $i$ with the ground-truth structure token $\boldsymbol{z}_i^{(0)}$, the Denoiser learns to estimate a Gaussian noise $\epsilon \in \mathbb{R}^{d_{\text{struct}}} \sim \mathcal{N}(0, \boldsymbol{I})$ based on three factors: the residue representation $\boldsymbol{c}_i$ containing its unmasked contextual information; a timestamp $t'$ randomly sampled from $\{1, 2, \ldots, T'\}$; and a noised token at the $t'$ step, formulated as $\sqrt{\bar{\alpha}_{t'}} \boldsymbol{z}_i^{(0)} + \sqrt{1 - \bar{\alpha}_{t'}} \epsilon$, where the $\bar{\alpha}_{t'}$ is defined by a noise scheduler (Ho et al., 2020; Nichol & Dhariwal, 2021).

Eventually, all learnable parameters are optimized through an overall objective:

$$\mathcal{L} = \mathcal{L}_{\text{struct}} + \gamma \mathcal{L}_{\text{seq}}, \quad (10)$$

where $\gamma$ balances the focus between protein sequence and structure modeling. Detailed settings of hyperparameters related to the loss are explained in D.2.

**Multimodal Protein Generation.** With the *per-token distribution* learned in parallel, the sequence and structure tracks employ different samplers, correspondingly. Taking residue $i$ with temporary condition representation $\boldsymbol{c}_i^{(t_s)}$ as an example, the masked sequence prediction can be done by the vanilla categorical sampler:

$$p(\hat{\boldsymbol{s}}_i^{(0)} | \boldsymbol{s}^{(t_s)}, \boldsymbol{z}^{(t_s)}) = \text{Softmax} \left( \text{Projector} \left( \boldsymbol{c}_i^{(t_s)} \right) / \tau_s \right), \quad (11)$$

where $\tau_s$ is the generation temperature for sequence. Meanwhile, given a hidden condition representation $\boldsymbol{c}_i^{(t_z)}$, the masked structure prediction undergoes a reverse diffusion procedure of DDPM (Ho et al., 2020), generating $\hat{\boldsymbol{z}}_i^{(0)}$ from a Gaussian noise over $T'$ steps:

$$\hat{\boldsymbol{z}}_i^{(t'-1)} = \frac{1}{\sqrt{\alpha_{t'}}} \left( \hat{\boldsymbol{z}}^{(t')} - \frac{1 - \alpha_{t'}}{\sqrt{1 - \bar{\alpha}_{t'}}} \text{Denoiser}(\hat{\boldsymbol{z}}_i^{(t')}, t', \boldsymbol{c}_i^{(t_z)}) \right) + (\sigma_{t'} \delta) \tau_z, \quad (12)$$

where $\tau_z$ controls the generation temperature for structure, $\delta$ is randomly sampled from the Gaussian distribution $\mathcal{N}(\boldsymbol{0}, \boldsymbol{I})$, and $\sigma_{t'}$ represents the noise level at denoising time step $t'$ (Li et al., 2024). The reverse procedure of DDPM naturally supports classifier-free guidance (CFG) (Ho & Salimans, 2022). In the context of multimodal protein modeling, we can consider the whole protein sequence track as the guidance condition, aiming to generate more self-consistent protein structure tokens.

To recap, the ***multimodal protein generation*** process follows the reverse diffusion language modeling process formulated in Equation 6, starting from a state where both sequence and structure are either fully masked (for unconditional sequence-structure co-generation) or partially masked (for motif-scaffolding, structure prediction, and inverse folding). For each step of the iterative generation, the model predicts all masked tokens, then selectively retains a certain proportion of these predictions while re-masking the remainder for the next step. Detailed procedures are also provided in the Appendix C.2-C.3.

## 4 EXPERIMENTS

We primarily evaluate HD-Prot models on four foundational tasks: unconditional protein sequence-structure co-generation (Section 4.1), motif-scaffolding (Section 4.2), protein structure prediction (Section 4.3), anad inverse folding (Section 4.4). Please refer to the Appendix D for implementation details, including the training dataset and training process of our model, the implementation of baseline models for all tasks, and the calculation process of evaluation metrics.

Table 2: Evaluation of Unconditional Protein Sequence-Structure Co-Generation. * denotes the performance of the MultiFlow variant (w/o data distillation) reported by Wang et al. (2024b).

| Models (#Params, #Training Sample) | Designability | | | Diversity | | Novelty | |
|---|---|---|---|---|---|---|---|
| | pLDDT ↑ | scRMSD ↓ | scTM ↑ | #CL@50 ↑ | #CL@95 ↑ | pdb-TM ↓ | sp-TM ↓ |
| MultiFlow (21M, 22.8K) | 79.271 ± 7.978 | 2.955 ± 4.252 | 0.937 ± 0.100 | 55.12 ± 15.79 | 100.00 ± 0.00 | 0.729 ± 0.135 | 0.730 ± 0.142 |
| * MultiFlow | 61.519 | 9.306 ± 8.499 | 0.750 ± 0.163 | 49.00 | - | - | - |
| ESM3 (1.4B, ∼1.08B) | 76.079 ± 13.53 | 31.98 ± 33.87 | 0.762 ± 0.221 | 48.00 ± 16.82 | 96.24 ± 7.704 | 0.862 ± 0.143 | 0.834 ± 0.213 |
| La-Proteina (160M, 550K) | 80.152 ± 10.51 | 4.477 ± 6.652 | 0.923 ± 0.141 | 64.32 ± 9.586 | 100.00 ± 0.00 | 0.715 ± 0.186 | 0.710 ± 0.188 |
| - w/ triangular layers | 83.770 ± 10.13 | 3.260 ± 6.317 | 0.953 ± 0.119 | 40.60 ± 22.45 | 100.00 ± 0.00 | 0.795 ± 0.182 | 0.784 ± 0.188 |
| DPLM-2 (150M, 220K) | 82.525 ± 7.754 | 5.125 ± 5.101 | 0.895 ± 0.112 | 43.28 ± 7.871 | 83.08 ± 8.665 | 0.918 ± 0.058 | 0.932 ± 0.054 |
| DPLM-2 (650M, 220K) | 81.920 ± 8.643 | 4.899 ± 5.523 | 0.906 ± 0.105 | 52.40 ± 6.083 | 82.40 ± 8.765 | 0.920 ± 0.073 | 0.933 ± 0.069 |
| DPLM-2.1 (650M, -) | 84.773 ± 7.719 | 5.076 ± 5.155 | 0.898 ± 0.114 | 60.40 ± 5.766 | 89.28 ± 6.059 | 0.898 ± 0.102 | 0.929 ± 0.068 |
| **HD-Prot** (155M, 210K) | 80.646 ± 11.07 | 4.629 ± 4.709 | 0.887 ± 0.127 | 44.32 ± 7.409 | 78.32 ± 12.84 | 0.895 ± 0.119 | 0.914 ± 0.114 |
| **HD-Prot** (670M, 210K) | 81.099 ± 9.832 | 4.899 ± 4.534 | 0.878 ± 0.126 | 51.16 ± 6.593 | 86.08 ± 4.672 | 0.897 ± 0.103 | 0.918 ± 0.095 |
| PDB Proteins | 79.075 ± 13.03 | 4.669 ± 7.683 | 0.905 ± 0.143 | 55.80 ± 5.671 | 78.40 ± 3.499 | - | - |

## 4.1 Unconditional Protein Sequence-Structure Co-Generation

In this task, models are required to generate proteins with both sequences and structures simultaneously, using only the specified protein length as input. We compare our HD-Prot model with one state-of-the-art protein co-generation method, i.e., La-Proteina (Geffner et al., 2025a), and four multimodal protein models, i.e., MultiFlow (Campbell et al., 2024), ESM3 (Hayes et al., 2025), DPLM-2 (Wang et al., 2024b), and DPLM-2.1 (Hsieh et al., 2025). Specifying the protein length as 100, 200, 300, 400, and 500, we employ each method to generate 100 proteins and repeat for five different seeds, respectively. Moreover, $5 \times 100$ distinct PDB proteins are randomly selected to serve as reference samples.

**Quantitative Analysis.** Referring to Campbell et al. (2024) and Wang et al. (2024b), the generation results are quantitatively evaluated by three sets of metrics, namely the designability, diversity, and novelty. (1) **Designability**. A generated protein is considered designable if its sequence is foldable and its structure is consistent with the sequence's structure prediction result. The *foldability* of a generated sequence is assessed using the pLDDT score given by ESMFold (Lin et al., 2023) during structure prediction. Meanwhile, *self-consistency* between the co-generated structure and the ESMFold-predicted structure is evaluated using backbone scRMSD and scTM. (2) **Diversity**. For a set of generated proteins, we calculate the number of clusters derived by Foldseek (Van Kempen et al., 2024) with the TM-score threshold at 0.5 and 0.95, resulting in #Cluster@50 and #Cluster@95. (3) **Novelty**. A generated protein is novel if it is dissimilar to well-known proteins, e.g., the PDB (wwp, 2019) or AlphaFoldDB-SwissProt (Jumper et al., 2021) proteins. We search for the most similar protein in a reference database, leading to pdb-TM and sp-TM.

Table 2 and Appendix Figure 6 shows the overall comparison results. Notably, natural proteins with absolute self-consistency still do not achieve perfect scores on these metrics, despite their strong overall performance. This can be somehow attributed to the use of ESMFold's predicted structure as a reference, which introduces a certain level of model bias. Therefore, we consider the performance of natural proteins as a special baseline: if a model surpasses this baseline, it may suggest an idealized outcome. For example, MultiFlow, enhanced with data distillation, substantially outperforms other models as well as the natural protein baseline (i.e., PDB proteins) in designability, diversity, and novelty. However, this may be because the model fits the simplified distribution of the distilled data instead of learning the more complex original protein knowledge (Campbell et al., 2024; Wang et al., 2024b). When the data samples distilled by ProteinMPNN are removed, MultiFlow's performance degrades substantially, particularly collapsing in its sequence generation ability. Additionally, La-Proteina (Geffner et al., 2025a) achieves state-of-the-art performance by being sufficiently scaled up with great computational efforts on a carefully curated subset of representative proteins from the AlphaFold database.

In pLMs that generally have more solid sequence modeling ability, ESM3 is significantly lagging behind. Although ESM3 undergoes extensive pre-training of masked language modeling across dynamic mask rates, it still struggles with prediction under high mask rates, resulting in suboptimal performance in unconditional sequence–structure co-generation. In contrast, the DPLM families show the state-of-the-art performance. The proteins they generate exhibit self-consistency and diversity similar to that of natural proteins, despite the cost of novelty. More importantly, our HD-Prot models exhibit competitive performance with the DPLM families. HD-Prot (155M) presents a high degree of designability common to that of DPLM-2 (150M); Meanwhile, all HD-Prot (670M), DPLM-2 (650M), and DPLM-2.1 (650M) models show a similar trend of enhancing the diversity of

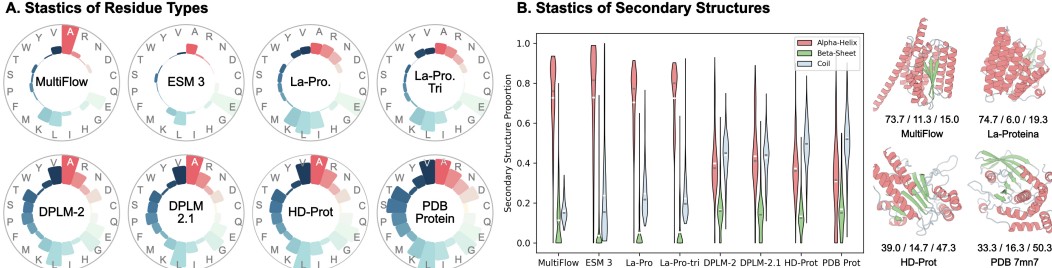

Figure 3: Qualitative Analysis. (**A-B**) The proteins generated by HD-Prot exhibit a similar distribution of residue types and secondary structure proportions compared to native proteins.

generation as the scale of parameters grows. Moreover, an interesting observation is that our HD-Prot performs better in the scRMSD compared to the scTM, i.e., excels more in the generation of local structure details. This advantage may stem from the use of continuous structure tokens, which capture fine-grained residue-level conformational details more accurately.

Furthermore, it is worth noting that the HD-Prot framework can be implemented with great computational efficiency. Due to the limitations of computational resources, we make many compromises in the implementation, especially the pre-cached tokenization results, mixed-precision training, and a smaller batch size. Ultimately, our HD-Prot model can be successfully trained on only one GPU. If converting the number × device × days into the rental price, our training cost is less than ***one twentieth*** of that of DPLM-2 (explained in the Appendix E.2), ensuring a fair comparison as both models are built upon the same foundation model and use similar training data.

**Qualitative Analysis.** Figure 3 shows the assessments of protein samples generated by each method. On the sequence modality, we compute the amino acid frequencies and visualize the distributions using Nightingale rose charts in Figure 3.A. The sequences generated by MultiFlow contain an unusually large amount of Alanine (A), and the categorical distribution learned by ESM3 and La-Proteina is biased toward Glutamic (E). In contrast, DPLM-2 series and HD-Prot models can generate protein sequences with a relatively balanced ratio of various amino acids, similar to natural proteins. On the structure modality, statistics of the proportion of secondary structures are presented in Figure 3.B. MultiFlow, ESM3 and La-Proteina exhibit a strong bias toward generating alpha-helices over beta-sheets and coils, whereas DPLM-2 series and HD-Prot produce proteins with secondary structure distributions that more closely resemble natural compositions. We select case samples with a length of 300 residues and a secondary structure ratio that is close to the corresponding average values. It is observed that structures generated by MultiFlow usually look ordered, with one clump after another of alpha-helix or beta-sheets, and very few coils. However, native structures and HD-Prot-generated structures could contain nearly half of them as coils, therefore looking more "flexible".

We attribute the similar unconditional generation performance of DPLM-2&-2.1 and our HD-Prot model to the closely aligned training datasets (Appendix D.1). These results indicate that, for building multimodal protein models upon sequence-based pLMs, quantization-based tokenization of structures is not the only viable path. Effectively integrating continuous structural representations into pLMs offers an alternative route that also successfully captures the underlying data distribution. Furthermore, case studies can be found in the Appendix E.4, including visualizations of some excellent sequence-structure co-generation results, and an analysis of the typical failure mode.

**Ablation Study.** Among various factors related to the implementation of HD-Prot, we identify three key findings, with experimental results presented in Table 3. First, *the pre-trained sequence foundation model is of great significance.* As shown in row 1, when training from scratch, our current data (∼210K proteins) remains insufficient to support effective language modeling, even for pLM at a relatively small scale of 150M parameters.

Second, we need to skillfully control the scale of fine-tuning, achieving *a balance between retaining sequence knowledge and acquiring more structure knowledge.* On the one hand, while performing modal extension based on a sequence-only pLM, HD-Prot faces the challenge of original modal collapse, which is not that severe but does exist. A 150M-parameter pLM can perform full-model fine-tuning without observing noticeable degradation in the co-generated sequence

Table 3: Ablation Study. #Param indicates tunable params (total pLM params) + denoising head params.

| | FM | #Param (M) | CFG | pLDDT ↑ | scRMSD ↓ | #CL@50 ↑ |
|---|---|---|---|---|---|---|
| 1 | × | 150 (150) + 5 | - | 73.100 | 6.798 | - |
| 2 | ✓ | 32 (150) + 5 | ✓ | 81.520 | 4.580 | 39.610 |
| 3 | ✓ | 150 (150) + 5 | × | 80.155 | 4.804 | 42.040 |
| 4 | ✓ | 150 (150) + 5 | ✓ | 80.646 | 4.629 | 44.320 |
| 5 | ✓ | 74 (650) + 20 | × | 80.132 | 5.084 | 48.680 |
| 6 | ✓ | 74 (650) + 20 | ✓ | 81.099 | 4.899 | 51.160 |
| 7 | ✓ | 650 (650) + 20 | - | 73.455 | 6.970 | - |

quality (rows 3, 4). However, for a 650M-parameter pLM that can quickly fit our limited data, full-model fine-tuning leads to forgetting of sequence knowledge (row 7). On the other hand, in the absence of sequence modal collapse, scaling up the training enables the model to capture a broader data distribution and generate more diverse proteins. Using a larger foundation model, or an expanding number of learnable parameters through LoRA can all achieve this effect (rows 2, 4, 6).

Third, *classifier-free guidance (CFG) (Ho & Salimans, 2022) can help generate high-quality continuous structure tokens*. Breaking down the generation process of HD-Prot into each step, the unconditional sequence-structure co-generation can actually be regarded as iterative per-token sampling under cross-modal conditioning. When generating a certain structure token, replacing the sequence track with masks is essentially performing a special "unconditional" generation. Therefore, we can employ the classic classifier-free guidance — steering the sampling of continuous structure tokens towards better consistency by combining the "conditional" and "unconditional" predictions. It is observed that employing CFG improves sequence-structure consistency without impairing generation diversity (rows 3-4, 6-7). Appendix E.5 provides ablations of the combined effects of two main sampling hyperparameters, i.e., the generation temperature of structure tokens and the CFG scale.

## 4.2 MOTIF-CONDITIONED PROTEIN SEQUENCE-STRUCTURE CO-GENERATION

Motif refers to a significant local pattern within a protein, while scaffold denotes the overall global structural framework that supports these motifs. Motif-scaffolding aims to design a stable protein scaffold that correctly positions one or more specified motifs. We adopt the experimental setup of Yim et al. (2024) and Wang et al. (2024b) across 24 motif-scaffolding tasks, sampling 100 scaffolds for each task in a run. The scaffold length and motif order are determined according to specifications. While focusing on the sequence-structure co-generation, both the sequence and structure of

Table 4: Motif-Scaffolding Results. #Solved presents "mean (min, max)" problems solved over repeats. * results are quoted from Wang et al. (2024b).

|  | #Solved / 24 | Avg. Success |
|---|---|---|
| * ESM3 | 20 | 17.58% |
| DPLM-2 (150M) | 15.6 (14, 17) | 20.0% ± 7.0% |
| DPLM-2 (650M) | 17.8 (16, 19) | 27.7% ± 0.8% |
| HD-Prot (155M) | 18.2 (18, 19) | 15.9% ± 0.3% |
| HD-Prot (670M) | 19.4 (19, 21) | 24.1% ± 1.1% |

the motif are provided as the input condition. A motif-scaffolding case is considered successful if it meets the requirements of overall designability and local motif-preserving at a time. Specifically, the criteria require scTM > 0.8 and motif-RMSD < 1.0 Å (Wang et al., 2024b), ensuring both self-consistency between the predicted structure of the generated sequence and the directly generated structure, as well as accuracy in the predicted motif structure relative to the native motif.

We evaluate HD-Prot against ESM3 and DPLM-2 based on the number of solved problems and success rate. Table 4 and Appendix E.3 summarize the results of five repetitions of sampling with different random seeds. The results demonstrate that HD-Prot effectively generates scaffolds that precisely match the given motifs. It successfully solves at most 21 out of 24 sub-tasks, outperforming both ESM3 and DPLM-2. Additionally, HD-Prot achieves a comparable average success rate, with approximately one-quarter of all $24 * 100$ generated samples meeting the success criteria. These results, while preliminary, underscore the potential of sequence-structure co-generation as an effective strategy for advancing conditional protein design. Besides, an analysis of the sampling hyperparameters of HD-Prot can be found in Appendix E.5.

## 4.3 PROTEIN STRUCTURE PREDICTION

Protein structure prediction aims to infer the 3D structure of a protein according to its amino acid sequence (Jumper et al., 2021; Lin et al., 2023). In the context of sequence-structure joint-modeling, protein structure prediction is also considered a sequence-conditioned structure generation task. Following the experimental setup of Wang et al. (2024b); Hsieh et al. (2025), we evaluate the protein structure

Table 5: Evaluation of Protein Structure Prediction. * results are quoted from Wang et al. (2024b).

| Model | CAMEO | | PDB Date Split | |
|---|---|---|---|---|
|  | RMSD ↓ | TM-score ↑ | RMSD ↓ | TM-score ↑ |
| * MultiFlow | 17.840 ± 17.96 | 0.810 ± 0.880 | 15.640 ± 16.08 | 0.530 ± 0.490 |
| ESM3 | 5.377 ± 6.303 | 0.860 ± 0.168 | 4.042 ± 4.824 | 0.883 ± 0.150 |
| DPLM-2 (150M) | 9.919 ± 6.994 | 0.720 ± 0.189 | 7.833 ± 6.004 | 0.765 ± 0.169 |
| DPLM-2 (650M) | 7.483 ± 6.126 | 0.786 ± 0.170 | 5.253 ± 5.143 | 0.836 ± 0.144 |
| DPLM-2.1 | 6.272 ± 6.202 | 0.824 ± 0.166 | 2.869 ± 3.942 | 0.915 ± 0.113 |
| HD-Prot (155M) | 9.185 ± 6.316 | 0.719 ± 0.201 | 6.229 ± 5.391 | 0.781 ± 0.181 |
| HD-Prot (670M) | 7.468 ± 6.004 | 0.769 ± 0.177 | 5.001 ± 4.565 | 0.827 ± 0.153 |

prediction capability of multimodal protein generative models via two datasets, i.e., CAMEO 2022, and a PDB Date Split curated by Campbell et al. (2024). The structure prediction results are compared to the corresponding native structures, and the `RMSD` and `TM-score` are calculated to assess the prediction accuracy.

Table 5 presents the comparison between HD-Prot and four multimodal protein generative models, where all predictions with randomness are repeated five times with different seeds. Firstly, compared with the unconditional protein sequence-structure co-generation results (Table 2), MultiFlow and ESM3 present totally different capabilities in protein structure prediction. Due to the reliance on non-natural distillation data, MultiFlow lacks the ability to understand the natural sequence arrangement as well as the sequence-structure folding rules. Meanwhile, given the complete sequence information, the ultra-large-scale pre-trained ESM3 model can accurately infer the corresponding structural information. Notably, HD-Prot performs better or comparable to the DPLM-2 at both ∼150M and ∼650M scales. During the training of HD-Prot, it has never seen a situation where the sequence track is completely given, and the structure track is fully masked. This absolutely zero-shot structure prediction performance indicates that HD-Prot has acquired considerable sequence-structure cross-modal capabilities. Besides, the explanation of the sampling hyperparameters of HD-Prot can be found in Appendix E.5.

### 4.4 INVERSE FOLDING

Inverse folding is also named the structure-conditioned sequence design, aiming to discover protein sequences that can fold into the given structures (Dauparas et al., 2022; Hsu et al., 2022). Referring to the experimental setup in Wang et al. (2024b); Hsieh et al. (2025), the CAMEO 2022 and PDB Date Split datasets are used for evaluation. Compared to the one-to-one structure prediction, the inverse folding has a one-to-many na-

Table 6: Evaluation of Inverse Folding. * results are quoted from Wang et al. (2024b).

| Model | CAMEO | | PDB Date Split | |
|---|---|---|---|---|
| | scRMSD ↓ | scTM ↑ | scRMSD ↓ | scTM ↑ |
| * MultiFlow | - | 0.870 ± 0.940 | - | 0.940 ± 0.960 |
| ESM3 | 3.944 ± 4.964 | 0.901 ± 0.141 | 2.262 ± 3.090 | 0.940 ± 0.093 |
| DPLM-2 (150M) | 5.999 ± 7.469 | 0.848 ± 0.175 | 4.002 ± 4.700 | 0.895 ± 0.126 |
| DPLM-2 (650M) | 4.659 ± 4.875 | 0.871 ± 0.154 | 3.114 ± 4.034 | 0.911 ± 0.113 |
| DPLM-2.1 | 4.304 ± 4.586 | 0.876 ± 0.141 | 2.271 ± 3.606 | 0.927 ± 0.112 |
| HD-Prot (155M) | 4.637 ± 4.730 | 0.863 ± 0.156 | 2.903 ± 3.683 | 0.919 ± 0.107 |
| HD-Prot (670M) | 4.675 ± 4.930 | 0.866 ± 0.151 | 2.871 ± 3.599 | 0.920 ± 0.103 |

ture. There could be multiple distinct amino acid sequences that can fold into a target structure, in addition to its natural sequence. Therefore, rather than calculating the recovery rate of the natural sequence, the evaluation should estimate the self-consistency between the target structure and refolded structure of the designed sequence (Liu et al., 2025). We calculate the `scRMSD` and `scTM` with the assistance of ESMFold (Lin et al., 2023).

The performance of HD-Prot and four baseline methods are summarized in Table 6, with all sampling procedures run five times with different seeds. The evaluation conclusions for each model are relatively close to those in Table 5. ESM3 stands out the best among all methods, excels in completing the remaining multimodal context when sufficient initial information is provided. Then, HD-Prot performs highly comparable to the DPLM-2 series at both ∼150M and ∼650M scales. Such completely zero-shot inverse folding results demonstrate that HD-Prot has estimated the sequence-structure joint-distribution sufficiently well. Besides, the sampling strategy of HD-Prot is analyzed in Appendix E.5.

## 5 CONCLUSION

Protein language models (pLMs) have recently become a popular solution for joint-modeling of protein sequence-structure. However, the majority of existing methods still suffer from the quantized representations of structural information. To this end, we propose a hybrid diffusion protein language model (HD-Prot), which enables the discrete protein language models to understand and generate continuous protein structure information. The model bridges the discrete-continuous modality gap in multimodal protein modeling and demonstrates the promising potential of using continuous structure tokens within pLMs. Extensive quantitative and qualitative experiments show that HD-Prot achieves competitive multimodal protein modeling performance compared to state-of-the-art multimodal pLMs, while requiring fewer computational resources for development.

## ETHICS STATEMENT

This study did not involve human participants or animals. All data used were obtained from publicly available sources, and no ethical approval was required. The authors declare no conflicts of interest.

## REPRODUCIBILITY STATEMENT

This manuscript (Sections 3 4), together with its Appendices C D E, provides a complete description of the proposed method and experimental setups, which is sufficient to understand and reproduce all the results. To further facilitate reproducibility, we have included the core source code as supplementary material ("hd-prot.zip"). Any additional details are available upon request. A full public release of the codebase with complete reproduction guidelines is planned upon publication.

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

**Table of Appendix:**

## A   LLM USAGE DISCLOSURE

The authors utilized a large language model (LLM) solely as a tool to assist with the polishing and refinement of the writing in this paper. The model was used exclusively for improving grammatical fluency, sentence structure, and overall clarity of the manuscript. All ideation, theoretical development, empirical research, and technical conclusions remain entirely the work of the authors. The authors take full responsibility for all content generated by the LLM and presented in this work.

## B   ANALYSIS OF CONTINUOUS STRUCTURE TOKENS

On the CAMEO 2022 test set, the salad tokenizer (Jendrusch & Korbel, 2025) demonstrates excellent performance, achieving high-fidelity reconstruction with scRMSD < 1.0 Å for 187 out of 194 test structures. Figure 4A. presents a random case and a selected bad case, demonstrating the capability and characteristics of the tokenizer. It is observed that while the tokenizer achieves consistently accurate *local* reconstructions, it may misorient structural elements in disordered regions, thereby compromising the global performance.

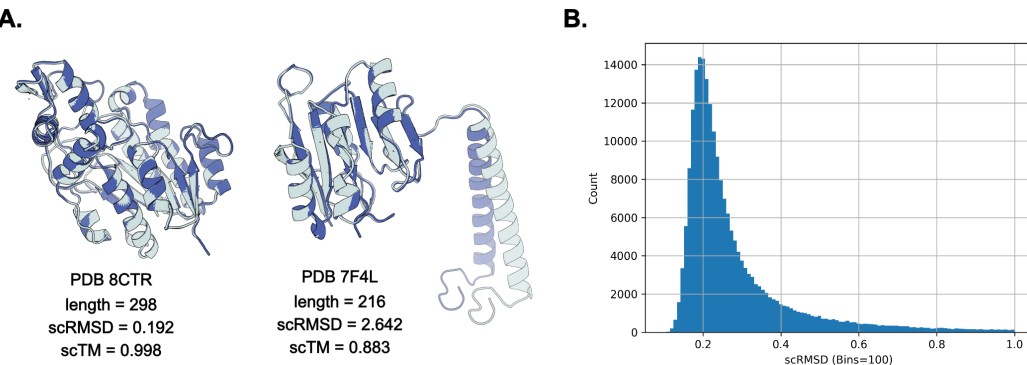

Figure 4: Analysis of Continuous Structure Tokens. (**A**) Visualization of protein structure reconstructions. (**B**) Statistics of the fidelity of continuous structure tokens.

As described in Section D.1, our training set contains approximately 210K proteins after various filterings. We pre-cache all those proteins into arrays of continuous structure tokens, and Figure 4B. presents the statistics of the structure reconstruction results based on these token arrays, reflecting their representational fidelity. The median scRMSD of 0.229 Å indicates excellent reconstruction quality, demonstrating that continuous structure tokens provide an extensively effective and nearly loss-free representation of protein structures.

While we keep the tokenizer frozen for computational efficiency, we have to adapt to its inherent numerical characteristics. Throughout our training dataset, the numerical mean value of continuous structure tokens is -0.432, and the variance is 28.562. In order to ensure the effective operation of the subsequent continuous diffusion learning based on Gaussian noise $\epsilon \sim \mathcal{N}(0, \boldsymbol{I})$, all continuous structure tokens undergo a very simple numerical scaling (Li et al., 2024). Using the statistical mean of the standard deviation as the scaling factor, the numerically divided tokens serve as ground truth for model learning, while the tokens generated by the model are scaled up accordingly for decoding by the tokenizer.

## C    FURTHER EXPLANATION OF HD-PROT

### C.1    MULTIMODAL PROTEIN MODELING

Previous studies (Campbell et al., 2024; Wang et al., 2024b) have shown that using decoupled sequence and structure diffusion schedulers enables multimodal protein models to accomplish comprehensive fundamental protein modeling. Table 7 summarizes the scheduler configurations and their corresponding protein modeling tasks. We denote the sequence scheduler as $t_s$ and the structure scheduler as $t_z$, where $t_s = 0$ or $t_z = 0$ represents the original clean data, and $t_s = T$ or $t_z = T$ corresponds to fully noised data.

Table 7: Scheduler Settings and Protein Modeling Tasks

|   | Sequence Scheduler | Structure Scheduler | Protein Modeling Task |
|---|---|---|---|
| 1 | $t_s \in \{0, 1, \ldots, T\}$ | $t_z = T$ | Sequence Generation |
| 2 | $t_s = T$ | $t_z \in \{0, 1, \ldots, T\}$ | Structure Generation |
| 3 | $t_s = 0$ | $t_z \in \{0, 1, \ldots, T\}$ | Protein Structure Prediction |
| 4 | $t_s \in \{0, 1, \ldots, T\}$ | $t_z = 0$ | Inverse Folding |
| 5 | $t_s = t_z \in \{0, 1, \ldots, T\}$ | | Sequence-Structure Co-Generation |

On the one hand, keeping one modality fully masked ensures independent generative modeling of the other modality. By configuring the schedulers as specified in rows 1 and 2 of Table 7, the model learns to perform protein sequence generation and protein structure generation, respectively. On the other hand, maintaining one modality fully visible drives the conditional generation of the other modality. The configuration in row 3 enables the model to learn sequence-conditioned structure generation, i.e., protein structure prediction. Similarly, the setting in row 4 facilitates structure-conditioned sequence generation, commonly known as inverse folding. Ultimately, by setting $t_s = t_z \in \{0, 1, \ldots, T\}$, the model learns sequence-structure dependencies across all possible masking ratios, thereby enhancing protein sequence-structure co-generation.

In the implementation of HD-Prot, we train our model with a combination of three scheduler settings, namely the sequence generation, structure generation, and sequence-structure co-generation. In each training batch, 20% of samples are treated with $t_s \in \{0, 1, \ldots, T\}$ and $t_z = T$ to help the pre-trained sequence-based pLM retain its sequence knowledge. Another 20% of samples are processed with $t_s = T$ and $t_z \in \{0, 1, \ldots, T\}$ to facilitate learning of the newly introduced protein structure modality. The remaining 60% of protein samples are processed with $t_s = t_z \in \{0, 1, \ldots, T\}$, enabling the model to learn the joint probability distribution of sequence and structure under positionally interlaced cross-modal conditioning. Interestingly, during the explicit training of protein sequence-structure co-generation, HD-Prot also implicitly learn to perform protein structure prediction and inverse folding. We hypothesize that the model, having learned the underlying principles of sequence-structure mapping at the token level, can apply them to complete tracks.

## C.2 MULTIMODAL PROTEIN GENERATION PROCEDURE

Firstly, we present the most basic procedure of ***unconditional protein sequence-structure co-generation*** in Algorithm 1. It primarily undergoes the reverse process of diffusion language modeling, i.e., iterative mask token prediction in parallel for both sequence and structure tracks. Concretely, in each iteration step, discrete sequence tokens are sampled from a categorical distribution, while continuous structure tokens are generated through the reverse process of Denoising Diffusion Probabilistic Model (DDPM) (Ho et al., 2020; Li et al., 2024).

---

**Algorithm 1:** Unconditional Protein Sequence-Structure Co-Generation.

**Input:**
- Network: Trained network $\theta = (\theta_b, \theta_s, \theta_z)$ (backbone, categorical head, denoising head)
- Hyperparams: Desired protein length $L$; Diffusion LM steps $T$
- Sequence track: Sampling temperature $\tau_s$
- Structure track: Sampling temperature $\tau_z$; DDPM steps $T'$; DDPM schedule $\beta_{t'}$

**Output:** Generated protein $(\boldsymbol{s}^{(0)}, \boldsymbol{z}^{(0)})$

1 **for** $i = 1, 2, \ldots, L$ **do**
2     $\boldsymbol{s}_i^{(T)} \leftarrow \boldsymbol{m}_s, \boldsymbol{z}_i^{(T)} \leftarrow \boldsymbol{m}_z$;          ▷ Initialize all tokens with masks
3 **end**

4 $k \leftarrow \lfloor L/T \rfloor$;          ▷ Number of tokens to update in each of the following steps

5 **Reverse Process of Diffusion Language Modeling:**

6 **for** $t = T, \ldots, 1$ **do**
7     $\boldsymbol{c}^{(t)} \leftarrow f_{\theta_b}(\boldsymbol{s}^{(t)}, \boldsymbol{z}^{(t)})$;          ▷ Inference through the main body of pLM
8     **Sequence Track Update:**
9     $\hat{\boldsymbol{s}}^{(0)} \sim \text{Softmax}(f_{\theta_s}(\boldsymbol{c}^{(t)})/\tau_s)$;          ▷ Sample sequence tokens from a categorical distribution
10     $\mathcal{I}_s^{(t)} \leftarrow \text{RandomSelect}\left(k, \{i \mid \boldsymbol{s}_i^{(t)} = \boldsymbol{m}_s\}\right)$;          ▷ Randomly select $k$ *masked* tokens to update
11     **for** $i = 1$ **to** $L$ **do**
12        **if** $i \in \mathcal{I}_s^{(t)}$ **then**
13           $\boldsymbol{s}_i^{(t-1)} \leftarrow \hat{\boldsymbol{s}}_i^{(0)}$;          ▷ Update with newly sampled sequence token
14        **else**
15           $\boldsymbol{s}_i^{(t-1)} \leftarrow \boldsymbol{m}_s$;          ▷ Keep the other sequence tokens masked
16        **end**
17     **end**
18     **Structure Track Update:**
19     $\hat{\boldsymbol{z}}^{(T')} \sim \mathcal{N}(0, \boldsymbol{I})$;          ▷ Sample continuous structure tokens starting from Gaussian noise
20     **for** $t' = T', T'-1, \ldots, 1$ **do**
21        $\alpha_{t'} := 1 - \beta_{t'}, \quad \bar{\alpha}_{t'} := \Pi_{n=1}^{t'}\alpha_n, \quad \sigma^2 = \beta_{t'}, \quad \delta \sim \mathcal{N}(0, \boldsymbol{I})$;
22                 ▷ DDPM scheduling parameters and randomly-sampled noise
23        $\hat{\epsilon} \leftarrow \epsilon_{\theta_z}(\hat{\boldsymbol{z}}^{(t')}, t', \boldsymbol{c}^{(t)})$;          ▷ Noise prediction
24        $\hat{\boldsymbol{z}}^{(t'-1)} \leftarrow \frac{1}{\sqrt{\alpha_{t'}}}\left(\hat{\boldsymbol{z}}^{(t')} - \frac{1-\alpha_{t'}}{\sqrt{1-\bar{\alpha}_{t'}}}\hat{\epsilon}\right) + (\sigma_{t'}\delta)\tau_z$;          ▷ DDPM denoising step
25     **end**
26     $\mathcal{I}_{z^{(t)}} \leftarrow \text{RandomSelect}\left(k, \{i \mid \boldsymbol{z}_i^{(t)} = m_z\}\right)$;          ▷ Randomly select $k$ *masked* tokens to update
27     **for** $i = 1$ **to** $L$ **do**
28        **if** $i \in \mathcal{I}_z^{(t)}$ **then**
29           $\boldsymbol{z}_i^{(t-1)} \leftarrow \hat{\boldsymbol{z}}_i^{(0)}$;          ▷ Update with newly sampled structure tokens
30        **else**
31           $\boldsymbol{z}_i^{(t-1)} \leftarrow \boldsymbol{m}_z$;          ▷ Keep the other tokens structure masked
32        **end**
33     **end**
34 **end**
35 **return** $(\boldsymbol{s}^{(0)}, \boldsymbol{z}^{(0)})$;          ▷ Return the generated protein

---

In the actual implementation, many extensions can be made to this basic generation process. We adopt some strategies native to the foundation sequence DPLM Wang et al. (2024a). Concretely, during the sampling of sequence tokens (Algorithm 1 row 9), a resampling scheme is included to prevent the generation of a large proportion of repetitive amino acids. Meanwhile, in addition to the naive random unmasking (row 10), top-$k$ unmasking strategy selects $k$ tokens with the highest sampling probability score for unmasking. During the sampling of structure tokens, alongside the noise estimation (row 23), classifier-free guidance (CFG) (Ho & Salimans, 2022) is introduced to enhance the sequence-structure self-consistency, with detailed operations described in C.3. Eventually, all generated sequence and structure tokens are translated back to the residue types and 3D coordinates by the tokenizers. Among those sampling hyperparameters, we by default set the diffusion LM steps $T = L$, the sequence sampling temperature $\tau_s = 1.0$, structure DDPM steps $T' = 100$, and the DDPM schedule $\beta_{t'}$ as a linear schedule. Besides, the trade-off between the self-consistency and diversity of generation results is largely controlled by the structure sampling temperature $\tau_z$ and the CFG scale. For HD-Prot (155M), the most balanced setting is $\tau_z = 0.35$ and CFG scale $= 2.0$. For HD-Prot (670M), the best setting is $\tau_z = 0.65$ and CFG scale $= 2.0$.

In addition to unconditional sequence-structure co-generation, HD-Prot is also capable of conditional generation, i.e., ***motif-scaffolding***. It is only necessary to modify the initialization of the input. Different from rows 1-4 of Algorithm 1, we don't initialize all tokens with masks. Given a motif with a length of $l$, its sequence is directly mapped to the sequence tokens, and its structure is first processed into continuous structure tokens via the tokenizer. According to the specific motif position and scaffold length $L$ (Yim et al., 2024), the input consists of the sequence and structure tokens of the motif at their specific positions, while other positions are masked. HD-Prot gradually generates all mask tokens over $T = L - l$ steps, while maintaining the initialized motif tokens unchanged. After the final step, all sequence and structure tokens are translated back to the residue types and 3D coordinates by the tokenizers. For both HD-Prot (155M) and HD-Prot (670M), we by default set the sequence sampling temperature $\tau_s = 1.0$ and the structure sampling temperature $\tau_z = 0.1$, without using the classifier-free guidance.

To accomplish the ***protein structure prediction***, the sequence track of HD-Prot is initialized according to the given protein sequence, and the structure track is completely filled with mask tokens. No matter what the length of the given protein sequence is, HD-Prot predicts all structure tokens in one step, i.e., setting $T = 1$. By default, the structure sampling temperature $\tau_z = 0.0$, without employing the classifier-free guidance. Eventually, the generated continuous structure tokens are transformed into 3D coordinates via the structure tokenizer. Correspondingly, for ***inverse folding***, a given protein structure with length $L$ is firstly processed into the continuous structure tokens by our protein structure tokenizer. Then, the structure track of HD-Prot is initialized by those structure tokens, and the sequence track is set as fully masked. By default, HD-Prot gradually predicts all sequence tokens over $T = L$ steps with sequence sampling temperature $\tau_s = 0.1$. The finally obtained sequence tokens are directly mapped to the amino acid sequence.

Consistent with previous studies (Hayes et al., 2025; Wang et al., 2024b), we have identified different sampling strategies for different multimodal protein generation tasks. We provide further ablation studies in Section E.5. It is observed that the sampling strategy should match the strength of the given condition. With a stronger condition provided, the model is confronted with a narrower exploration space. Thus, it performs better at a lower temperature and with fewer diffusion steps, meanwhile eliminating the need for guidance.

## C.3 CLASSIFIER-FREE GUIDANCE FOR CONTINUOUS STRUCTURE TOKENS

Classifier-free guidance (Ho & Salimans, 2022) has been extensively utilized in diffusion generative models. For example, in vision models and vision language models, it is generally assumed that CFG is used to generate high-quality images that better match the condition labels or prompts (Li et al., 2024; Wu et al., 2025). The core idea of CFG is to extrapolate the model's output by combining a conditional prediction and an unconditional prediction, steering the generation towards the condition by increasing the scale of the difference between them. It concretely adjusts the noise estimation of diffusion models through:

$$\hat{\epsilon} \leftarrow (1 - \omega) \cdot \underbrace{\epsilon_\theta \left( \boldsymbol{x} \mid \emptyset \right)}_{\text{unconditional}} + \omega \cdot \underbrace{\epsilon_\theta \left( \boldsymbol{x} \mid \boldsymbol{c} \right)}_{\text{conditional}}, \tag{13}$$

where $x$ denotes the model's input general input content, $c$ denotes the generation condition, and $\omega$ represents the CFG scale.

Our HD-Prot framework fuses the sequence and structure information from the very beginning. Any change in the input sequence/structure is bound to have an impact on the output structure/sequence. Therefore, the unconditional sequence-structure co-generation process can be treated as $T$-step combination of cross-modal conditional generation of tokens. Specifically, we consider the whole sequence track as the condition for the sampling of continuous structure tokens, where fully masking the sequence track is a kind of "unconditional" case.

The DDPM generation process for continuous structure tokens naturally supports classifier-free guidance. Introducing the conditional and unconditional cases that we just explained, CFG changes the noise prediction described in row 23 of Algorithm 1, formally expressed as:

$$\hat{\epsilon} \leftarrow (1-\omega) \cdot \underbrace{\epsilon_{\theta_z}(\hat{\boldsymbol{z}}^{(t')}, t', \boldsymbol{c}_{\emptyset}^{(t)})}_{\text{unconditional}} + \omega \cdot \underbrace{\epsilon_{\theta_z}(\hat{\boldsymbol{z}}^{(t')}, t', \boldsymbol{c}^{(t)})}_{\text{conditional}}, \tag{14}$$

where $\boldsymbol{c}_{\emptyset}^{(t)} \leftarrow f_{\theta_b}(\emptyset, \boldsymbol{z}^{(t)})$ involves an additional inference through the pLM with sequence tokens all masked, and $\omega$ is the CFG scale.

# D  IMPLEMENTATION DETAILS

## D.1  TRAINING DATASET

A well-constructed training dataset plays an essential role in the successful training of protein generation models. Different "AI for Protein" projects have designed specific schemes to cluster and filter experimental and synthetic data from PDB (wwp, 2019) and AlphaFoldDB (Varadi et al., 2022). Our dataset is built referencing the DPLM-2 (Wang et al., 2024b), which utilizes approximately 20K PDB proteins and 200K APDB-Swissprot proteins. The former are representative clustering centers of PDB monomer proteins, and the latter are high-quality protein structure predictions with pLDDT $> 85$. Rather than directly using their data processing results, we independently obtain all protein structures based on the protein name list and perform additional filtering for structure reconstruction quality. We hypothesize that if a structure can not be excellently encoded and reconstructed by our protein structure tokenizer (Jendrusch & Korbel, 2025), it should be misleading to learn the probability distribution of the corresponding continuous structure tokens. By requiring the structure reconstruction quality with scRMSD $< 1.0$ and scTM $> 0.9$, our dataset ultimately includes 210,001 samples, i.e., 19,807 PDB proteins and 190,194 AFDB-SwissProt proteins.

As shown in Figure 5, the proteins in our training dataset have lengths ranging from 57 to 1024 residues. During the model's training, proteins longer than 512 residues are cropped to a length of 512. Furthermore, a random cropping strategy (Wang et al., 2024b) is also introduced to enhance data diversity. Any protein with more than 60 residues has a 50% chance of being cropped to a random length between 60 and its full length.

## D.2  TRAINING PROCESS OF HD-PROT

We employ the protein structure tokenizer with its pretrained parameters frozen. All training data are preprocessed into paired discrete sequence tokens and continuous structure tokens, then cached for efficient access. DPLM (Wang et al., 2024a), a pretrained sequence-based protein language model, is adopted as the foundation model. HD-Prot (155M) initializes its pLM backbone from DPLM (150M), and similarly, HD-Prot (670M) initializes from DPLM (650M). Overall, the trainable parameters include the pLM backbone (fine-tuned) and the remaining modules (trained from scratch). Based on our existing empirical observations, we recommend different training strategies for the two model scales: full-model fine-tuning for the 150M backbone, and a LoRA (Hu et al., 2022) configuration that yields $\sim$74M trainable parameters for the 650M backbone.

For hyperparameters, we adopt the reweighting scheme from Wang et al. (2024a) for the sequence track, setting $\lambda^{(t_s)} = 1 - (t_s - 1)/T$. For the structure track, we maintain a constant weight of $\lambda^{(t_z)} = 1$, following Li et al. (2024), and the DDPM diffusion schedule $\beta_{t'}$ is simply a linear schedule. The $\gamma$ used to combine the sequence/structure modeling losses is set as 0.2 empirically,

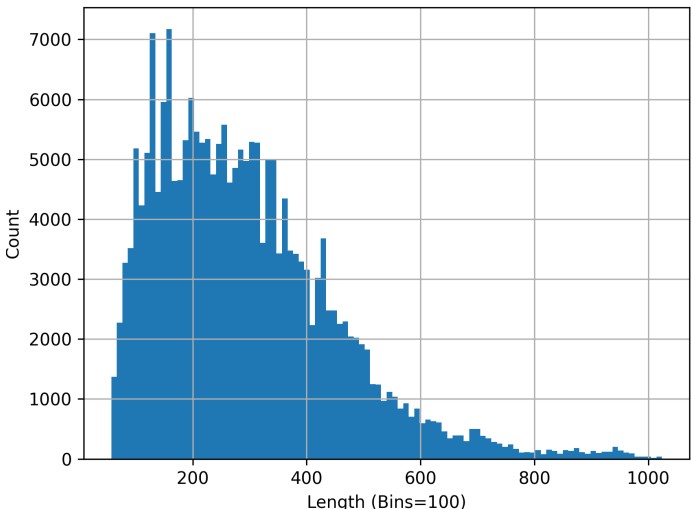

Figure 5: Length statistics of the training proteins

aiming to balance the magnitudes of the two loss values. For optimization, we use AdamW optimizer (Loshchilov & Hutter, 2017) with $\beta_1 = 0.9$, $\beta_2 = 0.95$ and the weight decay = 0.01. The mixed-precision technique is also introduced to reduce memory consumption. The training of HD-Prot (155M) runs for 120 epochs: warmup from 1e-5 to 1e-4 over the first 5 epochs, and linear decay to 1e-5 over the other 115 epochs. It takes **1** H20-8996G GPU for approximately 7 days. The training of HD-Prot (670M) runs for 60 epochs: warmup from 1e-5 to 1e-4 over the first 5 epochs, and linear decay to 1e-5 over the other 55 epochs. It takes **1** H20-96G GPU for about 10 days.

### D.3 IMPLEMENTATION OF BASELINE MODELS

For unconditional sequence–structure co-generation, we run MultiFlow (Campbell et al., 2024) and La-Proteina (Geffner et al., 2025a) using their official checkpoints and codebase (MultiFlow, La-Proteina). For La-Proteina, we evaluate both variants (with/without triangular updates) with the default noise scales of 0.1 for the alpha carbon atoms and 0.1 for the latent variables.

DPLM-2 series (Wang et al., 2024b; Hsieh et al., 2025) are also implemented by using their official checkpoints following the latest official instructions. For unconditional sequence-structure co-generation and motif-scaffolding, DPLM-2 uses default sampling strategies of "annealing@2.0:0.1" and "annealing@2.0:1.0", respectively, both over 500 steps. For protein structure prediction and inverse folding, DPLM-2 instead performs argmax sampling 100 steps. DPLM-2.1 by default adopts the "annealing@1.1:0.1" strategy for unconditional co-generation over 500 steps, and similarly uses argmax sampling for both protein structure prediction and inverse folding.

Notably, the ESM3 (Hayes et al., 2025) official provides the pre-trained checkpoint but has not specified how to perform unconditional sequence-structure co-generation. We adopt the suggestions of Yim et al. (2025) to perform a chain-of-thought inference to generate protein backbone structures first, including the sampling of secondary structure tokens with a temperature of 0.7, followed by the sampling of structure tokens with a temperature of 0.7. Subsequently, we sample the corresponding protein sequences at a temperature of 0.7. The three consecutive sets of sampling are all completed in $L$ steps ($L$ is the desired protein length). We attempted to implement ESM3 using the sequence-structure order instead of the secondary structure-structure-sequence order, or using other temperature settings, but did not achieve better results. According to the description in the original text, for protein structure prediction, ESM3 performs single-pass decoding with a temperature of 0.7. For inverse folding, ESM3 performs iterative decoding over $L/2$ steps using a constant temperature of 0.7.

## D.4 METRICS CALCULATIONS

Throughout all experiments, the `RMSD` and `TM-score` are calculated using standard functions in OpenFold (Ahdritz et al., 2024) and TM-Tools (Zhang & Skolnick, 2005).

The `#Clusters@50` and `#Cluster@95` are obtained by clustering the generated structures pooled by length with Foldseek (Van Kempen et al., 2024), using the following command:

```
foldseek easy-cluster ⟨input_path⟩ ⟨output_path⟩ ⟨tmp_path⟩
--alignment-type 1 --cov-mode 0 --min-seq-id 0 --tmscore-threshold {th},
```

where the tmscore-threshold is set as $th = 0.5$ or $th = 0.95$.

We quantify novelty by searching each generated protein against a reference database (PDB or AFDB-SwissProt) using Foldseek. The search is performed with the following command. The highest TM-score from the alignment against the PDB proteins is recorded as the `pdb-TM`, and that against AFDB-SwissProt proteins as the `sp-TM`.

```
foldseek easy-search ⟨input_path⟩ ⟨database_path⟩
⟨output_path⟩ ⟨tmp_path⟩ --exhaustive-search
--tmscore-threshold 0.0 --format-output query,target,alntmscore,
```

Notably, a hyperparameter `alignment-type` is by default set as 2 by foldseed, which means using the strategy of "3Di+AA Gotoh-Smith-Waterman" in searching for similar proteins. Aligning with our target of protein sequence-structure co-generation, this simultaneously measures the novelty of both the sequence and the structure of a protein.

## E FURTHER ANALYSIS OF EXPERIMENTAL RESULTS

### E.1 EVALUATION OF UNCONDITIONAL SEQUENCE-STRUCTURE CO-GENERATION

Figure 6 presents the detailed performance of a set of the DPLM-2 series and HD-Prot results grouped by the protein length, and shows the characteristics of native proteins for reference. The foldability, self-consistency, and diversity of native proteins are affected very little by changes in length. However, DPLM-2 series and HD-Prot all generate less consistent and more repetitive proteins when specifying a greater length. This seems to be a problem with the data. All natural proteins, irrespective of length, are governed by fundamental physical and evolutionary principles that underlie their stable existence. However, the principles have not been explicitly elucidated, and current AI models merely fit them implicitly in a data-driven manner. The limited presence of longer proteins in the training data (Figure 5) consequently leads to a drop in generation performance.

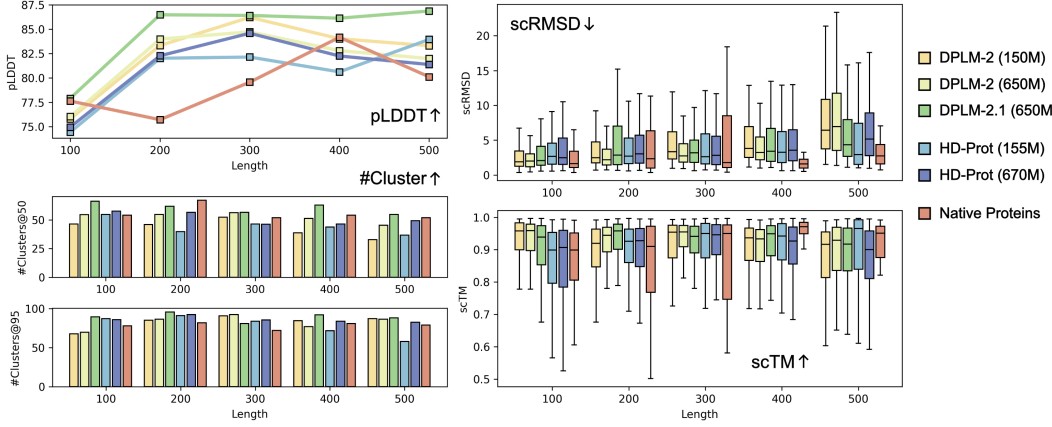

Figure 6: Evaluation on Unconditional Sequence-Structure Co-Generation

### E.2 EXPLANATION OF TRAINING COST

Table 8: Training Cost Comparison

| Model | Utilized GPU Type | Monthly Rental Price (CNY) | Training Duration (Count × Day) |
|---|---|---|---|
| DPLM-2 (150M)
DPLM-2 (650M) | Nvidia A100 | 19718 | 8 × 3
16 × 3 |
| HD-Prot (155M)
HD-Prot (670M) | Nvidia H20 | 4420 | 1 × 7
1 × 10 |

The computing resources required for developing HD-Prot and DPLM-2 are listed in Table 8.

Due to export control policies on the high-end AI accelerators, the Nvidia H20 GPU used for our experiments is only available in specific regions. Consequently, our cost estimation is based on the prevailing market rates of the primary cloud computing platform within the accessible region. On the AutoDL platform, renting one H20 (96G) GPU costs approximately 4420 CNY per month (148 CNY per day). Training the HD-Prot (670M) model requires 10 days on a single H20, leading to an estimated cost of *1480* CNY. For comparison, on the Volcengine platform, renting one A100 (80G) GPU costs about 19718 CNY per month (658 CNY per day). Training the DPLM-2 (650M) model, which required 16 A100 GPUs for 3 days, will cost approximately *31584* CNY. It suggests that our training cost is less than *one twentieth* of that of DPLM-2, at least within regions subject to GPU export restrictions.

### E.3 MOTIF-SCAFFOLDING RESULTS OF EACH PROBLEM

Table 9 details the motif-scaffolding results. Performance marked with * for ESM3 is reported by Wang et al. (2024b). For DPLM-2 and HD-Prot, we have conducted five repetitions using five different random seeds. We summarize the average, minimum, and maximum number of times each problem is solved, and report the average success rate with standard deviation.

### E.4 CO-GENERATION CASES & FAILURE MODE ANALYSIS

Fugure 7.A presents some excellent sequence-structure co-generation cases produced by HD-Prot. The selected samples with a length of 100-500 have a high degree of foldability ($\texttt{pLDDT} > 90$) and self-consistency ($\texttt{scRMSD} < 1.0$, $\texttt{scTM} > 0.9$). Meanwhile, although our model was trained primarily on proteins shorter than 512 residues, we can still find certain good cases for larger proteins with 600/700 residues.

Additionally, we select a typical poor generation case to analyze the failure mode, which is visualized in Figure 7.B. In this case, ESMFold yields a relatively high folding confidence globally, with average pLDDT $= 74.609$. However, in examining the pLDDT values assigned to each residue in detail, there is a short coil segment that connects a section of alpha-helix with the rest of the structure, which has lower pLDDT values ($70 > \text{pLDDT} > 50$). This indicates that ESMFold is uncertain about the exact orientation of the alpha-helix segment. We posit two plausible explanations: a less reasonable sequence generated by HD-Prot, or the presence of a biologically authentic disordered region. Then, the visualization of the protein structure alignment reveals that the alpha-helix segment is oriented in different directions in the co-generated structure and the predicted structure. This fundamentally led to a very poor RMSD score. Meanwhile, the alpha-helix in the co-generated structure is indeed of low quality, exhibiting distortions that do not conform to physical laws strictly. We posit that the corresponding continuous structure tokens remain noisy. Overall, we identify two common error patterns: 1) the structure orientation is misjudged when encountering unreasonable sequence fragments or disordered regions; 2) certain structure fragments are collapsed when the quality of the corresponding generated tokens is relatively low.

### E.5 ANALYSIS OF SAMPLING HYPERPARAMETERS

This section presents the ablation studies on critical sampling hyperparameters across all tasks. It has been identified that we need to adopt different sampling strategies according to the characteristics of various tasks. A task with low condition strength and a large solution space requires a

Table 9: Motif-Scaffolding Results of Each Problem

|  | * ESM3 | DPLM-2 (150M) | DPLM-2 (650M) | HD-Prot (155M) | HD-Prot (670M) |
|---|---|---|---|---|---|
| 1BCF | 23 | 6.4 (4, 10) | 0.8 (0, 2) | 5.4 (1, 9) | 9.6 (7, 14) |
| 1PRW | 54 | 88.8 (87, 91) | 80.2 (76, 85) | 70.4 (62, 79) | 78.8 (74, 82) |
| 1QJG | 3 | 0 | 0 | 0 | 0 |
| 1YCR | 18 | 29.2 (25, 33) | 38.2 (34, 46) | 44.2 (37, 53) | 45.2 (36, 61) |
| 2KL8 | 11 | 44.2 (39, 54) | 64.2 (58, 76) | 50.4 (46, 58) | 59.0 (55, 63) |
| 3IXT | 2 | 36.4 (32, 42) | 53.6 (44, 74) | 51.4 (48, 57) | 38.0 (33, 49) |
| 4JHW | 0 | 0 | 0 | 0 | 0 |
| 4ZYP | 8 | 4.8 (3, 6) | 11.6 (7, 15) | 0.4 (0, 1) | 2.0 (1, 3) |
| 5IUS | 0 | 0 | 0 | 0 | 0.2 (0, 1) |
| 5TPN | 1 | 0.4 (0, 1) | 0.4 (0, 1) | 15.2 (12, 20) | 11.8 (6, 15) |
| 5TRV_long | 19 | 2.2 (1, 5) | 1.6 (0, 3) | 8.6 (8, 10) | 8.6 (3, 13) |
| 5TRV_med | 16 | 6.2 (4, 10) | 6.6 (4, 9) | 11.4 (8, 15) | 20.0 (17, 25) |
| 5TRV_short | 1 | 0.8 (0, 2) | 1.6 (1, 3) | 10.4 (6, 16) | 17.4 (12, 23) |
| 5WN9 | 0 | 0.2 (0, 1) | 0 | 0 | 0.2 (0, 1) |
| 5YUI | 0 | 0 | 0 | 0 | 0 |
| 6E6R_long | 4 | 70.2 (68, 72) | 69.8 (65, 75) | 13.4 (9, 19) | 24.0 (18, 30) |
| 6E6R_med | 14 | 53.0 (50, 56) | 65.0 (61, 71) | 18.2 (16, 21) | 27.8 (25, 30) |
| 6E6R_short | 6 | 52.8 (50, 54) | 64.8 (62, 69) | 35.2 (28, 42) | 49.4 (37, 57) |
| 6EXZ_long | 13 | 30.6 (23, 37) | 53.6 (49, 60) | 6.6 (4, 10) | 36.0 (25, 39) |
| 6EXZ_med | 31 | 32.8 (30, 35) | 51.4 (46, 58) | 8.6 (5, 11) | 37.6 (28, 46) |
| 6EXZ_short | 28 | 20.0 (10, 24) | 28.8 (46, 58) | 12.2 (7, 16) | 52.0 (42, 61) |
| 7MRX_128 | 37 | 0 | 15.4 (6, 23) | 1.8 (0, 3) | 8.4 (5, 17) |
| 7MRX_60 | 59 | 0.6 (0, 2) | 30.4 (25, 38) | 9.2 (5, 13) | 31.6 (30, 33) |
| 7MRX_85 | 74 | 0 | 26.0 22, 32 | 7.4 (5, 11) | 21.6 (16, 26) |
| #Solved / 24 | 20 | 15.6 (14, 17) | 17.8 (16, 19) | 18.2 (18, 19) | 19.4 (19, 21) |
| Avg. Success | 17.58% | 20.0% ± 7.0% | **27.7%** ± 0.8% | 15.8% ± 0.3% | 24.1% ± 1.1% |

higher sampling temperature and a larger number of generation steps. In contrast, a task with a strong condition and a small solution space should better maintain a low temperature or even reduce the number of iterations. Notably, so far, our implementation of the classifier-free guidance is applicable only to the unconditional sequence-structure co-generation. For motif-scaffolding, the implicit assumption of conditioning the structure track on the sequence track is mismatched. For protein structure prediction, the solution space is sufficiently small that external guidance can even be counterproductive.

### E.5.1 UNCONDITIONAL SEQUENCE-STRUCTURE CO-GENERATION

For unconditional sequence-structure co-generation, the trade-off between the self-consistency and diversity of HD-Prot's outputs is largely controlled by two key factors: the structure sampling temperature $\tau_z$ and the classifier-free guidance scale. We have conducted a comprehensive grid search for both HD-Prot (155M) and HD-Prot (650M) over four temperature values and four CFG scales. Each configuration was evaluated with five random seeds, and we report the average performance across the following metrics: pLDDT, scRMSD, scTM, Inner-TM[1], #Cluster@50, and #Cluster@95. Figure 8 and Figure 9 provide a concise illustration of the effects in the form of heatmaps, where the darker the color, the better the metric.

Consistent with observations in other generative models, a higher sampling temperature increases diversity at the potential cost of sample quality. Empirically, we find that the fully fine-tuned model performs best with a lower temperature; hence, we set the default $\tau_z = 0.35$ for HD-Prot (155M). In contrast, for the LoRA-tuned model where the modality expansion is more constrained, a slightly higher temperature is beneficial, leading to our default choice of $\tau_z = 0.55$ for HD-Prot (670M).

Besides, with moderate strength of guidance (CFG scale = 2.0 or = 2.5), the pLDDT, scRMSD, and scTM metrics could be better than those without guidance (CFG scale = 1.0) or those with excessive guidance (CFG scale = 3.0). Meanwhile, a larger CFG scale generally leads to better

---

[1]Inner-TM is the average pairwise TM-score among the generated proteins with the same length.

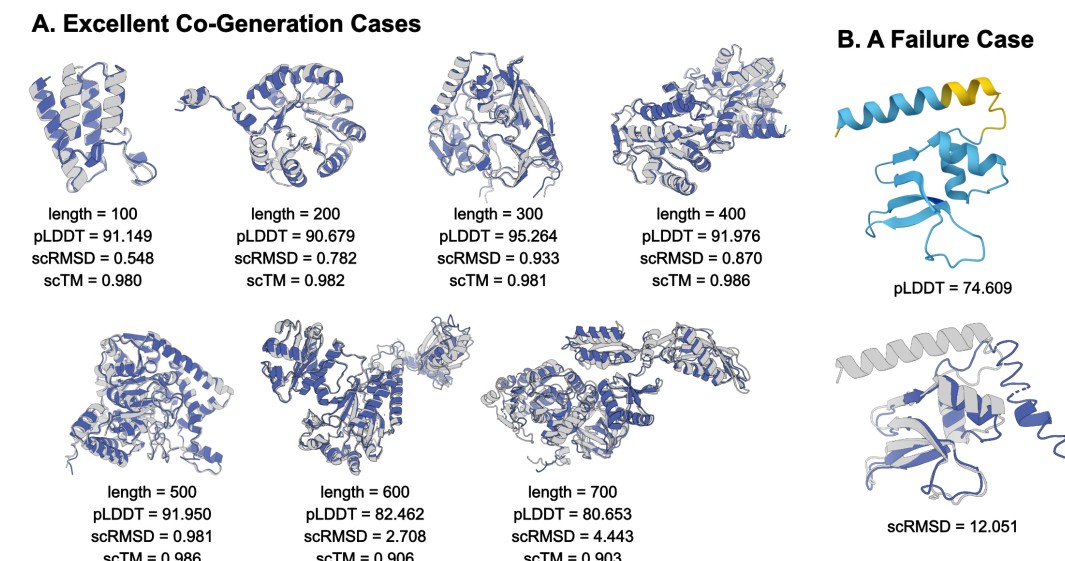

**A. Excellent Co-Generation Cases**

**B. A Failure Case**

length = 100
pLDDT = 91.149
scRMSD = 0.548
scTM = 0.980

length = 200
pLDDT = 90.679
scRMSD = 0.782
scTM = 0.982

length = 300
pLDDT = 95.264
scRMSD = 0.933
scTM = 0.981

length = 400
pLDDT = 91.976
scRMSD = 0.870
scTM = 0.986

pLDDT = 74.609

length = 500
pLDDT = 91.950
scRMSD = 0.981
scTM = 0.986

length = 600
pLDDT = 82.462
scRMSD = 2.708
scTM = 0.906

length = 700
pLDDT = 80.653
scRMSD = 4.443
scTM = 0.903

scRMSD = 12.051

Figure 7: Case study of HD-Prot's sequence-structure co-generation results. (**A**) Excellent Co-Generation Cases. In the structure alignment visualizations, the co-generated structures are colored blue, and the ESMFold-predicted structures are colored gray. (**B**) A Failure Case. The ESMFold-predicted structure is colored by the pLDDT scores, where the light blue indicates $90 > \texttt{pLDDT} > 70$ and the yellow indicates $70 > \texttt{pLDDT} > 50$. It is also aligned and compared to the co-generated structure.

diversity. Setting a medium CFG scale $= 2.0$ can relatively well balance the self-consistency and diversity of generated proteins.

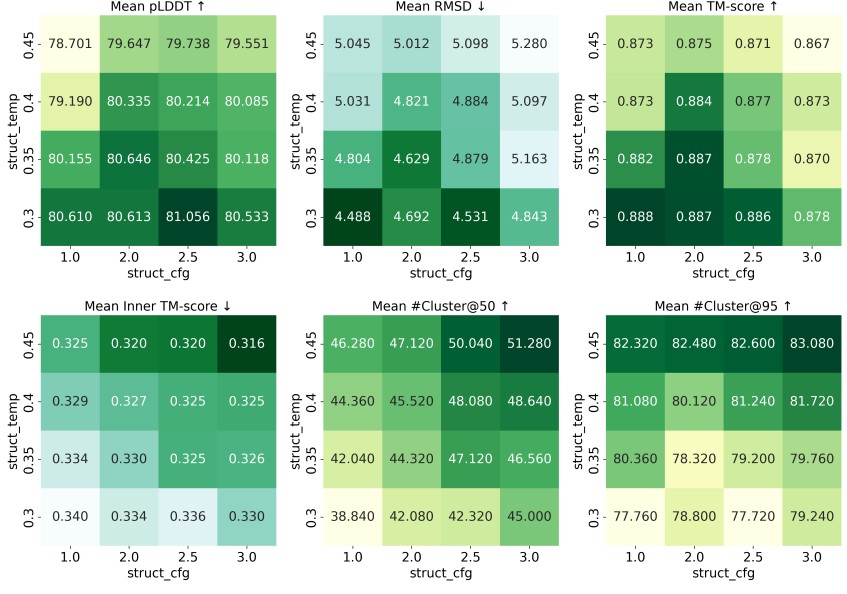

Figure 8: Unconditional Co-Generation Performance of HD-Prot (155M)

### E.5.2 MOTIF-SCAFFOLDING

Targeting a motif with a length of $l$ and a scaffold with a length of $L$, HD-Prot by default samples over $L - l$ steps to "complete" the tokens other than the initial motif tokens. The structure sampling

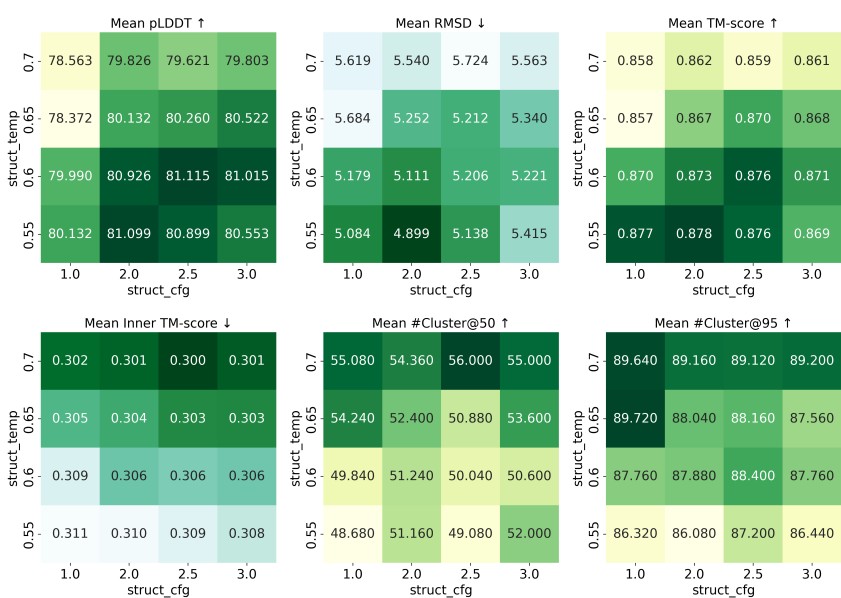

Figure 9: Unconditional Co-Generation Performance of HD-Prot (670M)

temperature is set as $\tau_z = 0.1$, and the classifier-free guidance is not introduced, which means the CFG scale $= 1.0$. To demonstrate the optimality of such default settings, Table 10 presents an ablation study by exploring three questions: 1) Should the structure sampling temperature be set at a higher level, as in the unconditional co-generation task, or should it be set relatively lower ($\tau_z = 0.35, 0.65 / 0.1$)? 2) Is the classifier-free guidance effective here (CFG scale $= 1.0 / 2.0$)? 3) Should we maintain the initialized motif tokens, or sample over $L$ steps to cover them (Maintain Init. Motif Tokens $=$ True / False)?

The answers can be drawn from the experimental results. First, using a lower sampling temperature can slightly increase the number of solved problems and the average success rate. Second, introducing classifier-free guidance always brings negative consequences. We suspect this is because the current CFG operation we have implemented is inconsistent with the objective of the Motif-Scaffolding task. As introduced in the Section C.3, CFG steers the generation of structure tokens to be more consistent with the sequence track at each step. However, motif-scaffolding requires both the final sequence and structure tracks to be more in line with the input Motif. Third, compared to sampling $L$ steps to cover the original motif tokens, sampling $L - l$ steps and retaining the initial motif tokens shows a slight advantage.

Table 10: Ablations on Motif-Scaffolding Performance of HD-Prot

| Model | Struct. Temp. | CFG Scale | Maintain Init. Motif Tokens | Avg. #Solved / 24 | Avg. Success |
|---|---|---|---|---|---|
| HD-Prot (155M) | 0.1 | 1.0 | True | **18.2** (18, **19**) | **15.9%** $\pm 0.3\%$ |
| | 0.1 | 2.0 | True | 17.8 (17, 18) | 15.1% $\pm 0.9\%$ |
| | 0.1 | 1.0 | False | **18.2** (18, **19**) | 15.8% $\pm 0.3\%$ |
| | 0.35 | 1.0 | True | 18 | 15.1% $\pm 0.7\%$ |
| | 0.35 | 2.0 | True | 17.6 (17, 18) | 14.2% $\pm 0.6\%$ |
| | 0.35 | 1.0 | False | 18 | 15.1% $\pm 0.7\%$ |
| HD-Prot (670M) | 0.1 | 1.0 | True | **19.4** (19, **21**) | **24.1%** $\pm 1.1\%$ |
| | 0.1 | 2.0 | True | 18.2 (18, 19) | 23.2% $\pm 0.5\%$ |
| | 0.1 | 1.0 | False | 18.8 (18, 19) | 23.6% $\pm 0.9\%$ |
| | 0.55 | 1.0 | True | 18.8 (18, 19) | 21.4% $\pm 0.6\%$ |
| | 0.55 | 2.0 | True | 18.2 (18, 19) | 20.9% $\pm 0.4\%$ |
| | 0.55 | 1.0 | False | 18.6 (18, 19) | 21.5% $\pm 0.6\%$ |

### E.5.3 PROTEIN STRUCTURE PREDICTION

Protein structure prediction is typically regarded as a near one-to-one mapping task. From the perspective of conditional generation, the input sequence acts as a highly restrictive condition, leaving only a narrow structural solution space. As shown in Table 11, which compares various sampling strategies, the optimal approach in this low-entropy regime is to set the structure sampling temperature to $0.0$ and perform generation in a single deterministic step. In contrast, increasing the sampling temperature, adding more iterative steps, or applying classifier-free guidance (CFG) introduces unnecessary stochasticity, which degrades structure prediction accuracy rather than improving it.

Table 11: Ablations on Protein Structure Prediction Performance of HD-Prot

| Model | Settings | | | CAMEO | | PDB Date Split | |
| --- | --- | --- | --- | --- | --- | --- | --- |
| | Temp. | CFG | T | RMSD | TM-score | RMSD | TM-score |
| HD-Prot (155M) | 0.0 | 1.0 | 1 | **9.199** ± 6.335 | **0.720** ± 0.200 | **6.231** ± 5.395 | **0.781** ± 0.181 |
| | 0.0 | 1.0 | L | 9.699 ± 6.621 | 0.713 ± 0.200 | 6.654 ± 5.685 | 0.776 ± 0.185 |
| | 0.1 | 1.0 | L | 9.716 ± 6.687 | 0.713 ± 0.200 | 6.653 ± 5.696 | 0.774 ± 0.188 |
| | 0.1 | 2.0 | L | 9.607 ± 6.501 | 0.711 ± 0.200 | 6.599 ± 5.663 | 0.772 ± 0.187 |
| | 0.35 | 1.0 | L | 9.734 ± 6.640 | 0.711 ± 0.200 | 6.648 ± 5.651 | 0.772 ± 0.187 |
| | 0.35 | 2.0 | L | 9.637 ± 6.448 | 0.709 ± 0.199 | 6.592 ± 5.581 | 0.770 ± 0.188 |
| HD-Prot (670M) | 0.0 | 1.0 | 1 | **7.468** ± 6.004 | 0.769 ± 0.177 | **5.001** ± 4.565 | 0.827 ± 0.153 |
| | 0.0 | 1.0 | L | 7.500 ± 5.982 | 0.776 ± 0.177 | 5.023 ± 4.780 | **0.832** ± 0.149 |
| | 0.1 | 1.0 | L | 7.525 ± 6.136 | **0.776** ± 0.176 | 5.014 ± 4.773 | 0.832 ± 0.150 |
| | 0.1 | 2.0 | L | 7.743 ± 6.181 | 0.767 ± 0.177 | 5.084 ± 4.711 | 0.825 ± 0.150 |
| | 0.55 | 1.0 | L | 7.757 ± 6.167 | 0.766 ± 0.177 | 5.065 ± 4.670 | 0.826 ± 0.150 |
| | 0.55 | 2.0 | L | 7.848 ± 6.451 | 0.759 ± 0.178 | 5.131 ± 4.700 | 0.820 ± 0.152 |

### E.5.4 INVERSE FOLDING

Inverse folding is usually considered a one-to-many prediction task. This task requires the generated sequence to adhere to the conditional structure, while allowing the exploration of diverse alternatives. Table 12 shows the comparison of the strategies for decoding each sequence token. It is observed that setting a small sampling temperature $\tau_z = 0.1$ is better than setting a larger temperature $\tau_z = 1.0$, and it is also better than directly using the deterministic argmax. That is to say, retaining few randomness is better than allowing excessive randomness, and it is also better than having no randomness at all.

Table 12: Ablations on Inverse Folding Performance of HD-Prot

| Model | Settings | | CAMEO | | PDB Date Split | |
| --- | --- | --- | --- | --- | --- | --- |
| | Strategy | Temp. | scRMSD | scTM | scRMSD | scTM |
| HD-Prot (155M) | Vanilla | 0.1 | 4.637 ± 4.730 | 0.863 ± 0.156 | 2.903 ± 3.683 | 0.919 ± 0.107 |
| | Vanilla | 1.0 | 4.689 ± 4.812 | 0.862 ± 0.150 | 2.928 ± 3.694 | 0.919 ± 0.106 |
| | Argmax | - | 4.830 ± 4.935 | 0.861 ± 0.151 | 2.872 ± 3.511 | 0.919 ± 0.104 |
| HD-Prot (670M) | Vanilla | 0.1 | 4.675 ± 4.930 | 0.866 ± 0.151 | 2.871 ± 3.599 | 0.920 ± 0.103 |
| | Vanilla | 1.0 | 4.750 ± 5.350 | 0.861 ± 0.152 | 2.944 ± 3.645 | 0.918 ± 0.103 |
| | Argmax | - | 4.708 ± 4.930 | 0.864 ± 0.146 | 2.900 ± 3.591 | 0.920 ± 0.103 |

