# OpenReview forum: "Empowering Protein Language Model for Sequence-Structure Co-Generation with Continuous Structure Tokens"
_ICLR.cc/2026/Conference — Submitted to ICLR 2026_

### Official Review · Reviewer_ai85 · 2025-10-31

**Soundness:** 1
**Presentation:** 3
**Contribution:** 1
**Rating:** 2
**Confidence:** 5

**Summary:**

This paper introduces HD-Prot, a hybrid diffusion protein language model that uses continuous (non-quantized) structure tokens instead of the discrete tokens employed by existing methods like ESM3 and DPLM-2. The approach extends sequence-based protein language models to jointly model discrete sequence tokens via categorical prediction and continuous structure tokens via diffusion modeling, unified through an absorbing diffusion process. The authors evaluate performance on protein sequence-structure co-generation and motif-scaffolding tasks, and provide ablation studies examining design choices including fine-tuning strategies and classifier-free guidance for continuous tokens.

**Strengths:**

* The information loss problems in existing discretization approaches are clearly articulated, providing motivation for exploring continuous structure tokens in protein language models.

* The paper is well-structured  and clearly presented.

* Table 3 provides valuable ablation findings on the importance of pre-trained foundation models, different fine-tuning strategies for different model scales, and the role of classifier-free guidance.

* The work successfully shows that continuous structure tokens can work within protein language models, opening a new research direction even if not demonstrably superior to existing methods.

**Weaknesses:**

* The work is fundamentally incomplete. Table 5 lists 5 protein modeling tasks that multimodal pLMs should handle (sequence generation, structure generation, structure prediction, inverse folding, co-generation), yet the paper evaluates only 2 tasks. This severely limits assessment of the approach's capabilities and fails to deliver on the promised comprehensive multimodal modeling.

* The small sample sizes undermine all conclusions. There is no justification provided for the inadequate size of the evaluation batch of 100 proteins. There are no error bars, confidence intervals, or significance tests. No multiple runs or random seeds reported. These numbers are far too small to draw reliable conclusions about model performance or make comparisons between methods.

* Missing obvious and critical baselines. The need for multimodality is not shown. Without proving superiority over simpler models, the value proposition of a complex multimodal architecture remains unsubstantiated. The authors should introduce basic baselines to directly test whether the complex co-generation setup provides any benefit over simpler unimodal approaches. For example, a basic pipeline like "Generate unconditional structure → ProteinMPNN (8×) → ESMFold → measure scRMSD/Diversity/Novelty". Also there are lots of unimodal models to compare to. Also, there is no comparison to the structure tokenizer alone: why build a language model on top if the tokenizer itself might generate comparable structures?

* The approach primarily adapts existing diffusion language modeling frameworks with continuous tokens. Given this limited novelty, the paper requires substantially more ablation studies and insights, yet critical hyperparameters remain completely unexplored. For example, there are no ablation of lambdas and gamma (Eqs. 8-10), no systematic analysis of τz (structure sampling temperature) despite claims it controls results, diffusion schedule parameters (T, T', βt') are completely unjustified.

* The evaluation metrics are poorly designed. #Cluster threshold=0.5 for the diversity  only detects highly different proteins; need e.g. 0.95 to detect mode collapse. Inner-TM is meaningless, the full distributions like Figure 6 should be drawn. The Novelty is measured against PDB/SwissProt instead of training set, so it doesn't really detect memorization. Length-dependent metric, scRMSD, averaged across different lengths obscures true performance. Also, Figure 3A-B compares to PDB, not the training set; different models trained on different data; sample size is not reported.

* Even with these metrics, the paper only demonstrates continuous tokens are viable, not superior. So, the claims of "highly competitive performance" are misleading.

* "Information loss" claims are fundamentally misleading, the paper conflates "continuous representation" with "no information loss". Compression from L×4×3 coordinates to L×dstruct is still lossy compression. "Continuous" does not equal "lossless",  this is dimensionality reduction regardless of discretization. By the way, the actual dstruct dimension not specified in main text.

* Motif-scaffolding evaluation is poor. The authors should have used the MotifBench benchmark, not the outdated RFDiffusion benchmark. The procedure description (line 819) vague: how exactly are motif tokens "maintained unchanged"? CFG improves consistency+diversity (Table 6) but not used here - why?

* The proposed approach is fundamentally unscalable. Table 3 reveals critical problems: training from scratch completely fails (row 1: pLDDT 73.100), full fine-tuning of the 650M model fails with sequence knowledge collapse (row 7: pLDDT 73.455), and the approach is fundamentally constrained to fine-tuning small pre-trained models with limited data. This severely limits future development and undermines the approach's viability.

* The proposed approach seems to be brittle and hyperparameter-sensitive. For example, different model sizes require completely different strategies (full fine-tuning vs. LoRA), different optimal temperatures needed (τz=0.35 vs 0.65), CFG required for decent performance.

* No failure mode analysis. Zero discussion of when/why approach fails, problematic protein types, or error patterns. Length-dependent degradation (Figure 6) attributed to "less training data" but never investigated.

* Generalization claims are unsubstantiated. Figure 3C shows 600-700 residue examples but no quantitative evaluation despite training limited to ≤512 residues.

**Questions:**

* What is the actual mutual information or reconstruction fidelity comparison between continuous structure tokens (L×dstruct) and discrete tokens (VQ-VAE codebooks)? Can you quantify the information preservation advantage?

* What happens when you encounter proteins that the salad tokenizer reconstructs poorly (scRMSD>1.0 or scTM<0.9)? Can HD-Prot handle these cases, or is it fundamentally limited by tokenizer quality?

* Why was classifier-free guidance not used in motif-scaffolding if it improves both self-consistency and diversity (Table 6)?

* In motif-scaffolding (line 819, Appendix C.2), how exactly are motif tokens "maintained unchanged" during the iterative generation process? Does this apply to both sequence and structure tracks simultaneously?

* Given that HD-Prot is built on DPLM, which is built on ESM-2, how much of the performance can be attributed to the foundation model versus your contributions?

---

> ### Author Response · Authors · 2025-12-03
>
> We sincerely thank the reviewer for thoughtful and detailed feedbacks. We greatly appreciate your recognition of several key aspects of our work, including our clear framing of the critical issue, the clarity and readability of our writing, and the practical empirical insights provided in our ablation studies. Most importantly, we deeply appreciate your acknowledgment: **our work successfully shows that continuous structure tokens can work within protein language models, opening a new research direction even if not demonstrably superior to existing methods**.
>
> We took the criticisms regarding evaluation completeness and statistical rigor very seriously. In this revision, we have:
> 1. **Added two new tasks**: Protein Structure Prediction and Inverse Folding (Sections 4.4, 4.5).
> 2. **Expanded statistical rigor**: Repeated experiments with 5 random seeds and reported standard deviations.
> 3. **Adjusted evaluation metrics**: Included #Cluster@95 and corrected metrics as suggested.
> 4. **Added failure mode analysis**: New Appendix E.4 discussing the method's limitations.
> 5. **Added sampling hyperparamter analysis**: New Appendix E.5 discussing the sampling strategies across the unconditional co-generation, motif-scaffolding, protein structure prediction, and inverse folding tasks.
>
> Our updated results consistently demonstrate that HD-Prot achieves performance competitive with state-of-the-art multimodal pLMs despite significantly lower computational costs, validating the viability of continuous structure tokens.

---

> ### Author Response · Authors · 2025-12-03
>
> > W1: Missing Evaluation Tasks
>
> We thank the reviewer for this valuable observation. We agree that a comprehensive multimodal protein language model should support a range of tasks, including not only unconditional sequence-structure co-generation and motif-scaffolding (the two core tasks we originally evaluated), but also modality-translation tasks such as protein structure prediction (sequence → structure) and inverse folding (structure → sequence).
> In direct response to this comment, **we have now extended our experiments to include both protein structure prediction and inverse folding**. We have carefully revised the paper to incorporate the two newly added tasks. Please refer to the text highlighted in blue in our updated PDF. Specifically:
> - We made minor adjustments to the wording in the **Abstract, Introduction, Preliminaries, and Conclusion** sections to better reflect the expanded scope.
> - In **Sections 4.4 and 4.5**, we present the evaluation setups and results for the protein structure prediction and inverse folding tasks.
> - In **Appendix C.2**, we provide a detailed description of HD-Prot's inference procedures for these two tasks.
> - In **Appendix D.3**, we detail how the baseline methods are implemented for these tasks.
> - In **Appendix E.5.3 and E.5.4**, we provide ablation studies on the sampling hyperparameters for these tasks.
>
> Here, we focus on discussing the main experimental results and new insights gained during our tuning of the sampling strategy.
>
> ### Protein Structure Prediction
>
> Following the experimental setup of DPLM-2 series [1] [2], we evaluate the protein structure prediction capability of multimodal protein generative models via two datasets, i.e., CAMEO 2022, and a PDB Date Split curated by MultiFlow [3]. The structure prediction results are compared to the corresponding native structures, and the RMSD and TM-score are calculated to assess the prediction accuracy.
>
> | Model                | CAMEO |       | PDB Date Split | |
> |----------------------|------------------|----------------|-------------------|----------------|
> |                      | RMSD ↓           | TM-score ↑     | RMSD ↓            | TM-score ↑     |
> | MultiFlow            | 17.840 ± 17.96   | 0.810 ± 0.880  | 15.640 ± 16.08    | 0.530 ± 0.490  |
> | ESM3          | 5.377 ± 6.303    | 0.860 ± 0.168  | 4.042 ± 4.824     | 0.883 ± 0.150  |
> | *DPLM-2 (150M)*    | 9.919 ± 6.994    | 0.720 ± 0.189  | 7.833 ± 6.004     | 0.765 ± 0.169  |
> | DPLM-2 (650M)        | 7.483 ± 6.126    | 0.786 ± 0.170  | 5.253 ± 5.143     | 0.836 ± 0.144  |
> | DPLM-2.1      | 6.272 ± 6.202    | 0.824 ± 0.166  | 2.869 ± 3.942     | 0.915 ± 0.113  |
> | *HD-Prot (155M)*   | 9.185 ± 6.316    | 0.719 ± 0.201  | 6.229 ± 5.391     | 0.781 ± 0.181  |
> | HD-Prot (670M)       | 7.468 ± 6.004    | 0.769 ± 0.177  | 5.001 ± 4.565     | 0.827 ± 0.153  |
>
> This table presents the statistical comparison between HD-Prot and four multimodal protein generative models, where all predictions with randomness are repeated five times with different seeds. The performance of the baseline methods on the Protein Structure Prediction aligns well with our observations in the Unconditional Sequence–Structure Co-Generation evaluation. ESM3 has undergone ultra-large-scale multimodal masked language modeling pretraining. However, its pretraining contains relatively few examples with high mask ratios, which limits its performance in unconditional co-generation. In the structure prediction task, where the full sequence is provided as input, ESM3’s conditional generation capability is fully leveraged, leading to strong performance. Meanwhile, the DPLM-2 series consistently present sufficiently good structure prediction performance, demonstrating mastery of cross-modal knowledge. Notably, HD-Prot performs better or comparable to the DPLM-2 at both ~150M and ~650M scales. During the training of HD-Prot, it has never seen a situation where the sequence track is completely given and the structure track is fully masked. This absolutely **zero-shot structure prediction performance indicates that HD-Prot has acquired considerably strong sequence-structure cross-modal capabilities**.

---

> ### Author Response · Authors · 2025-12-03
>
> To accomplish the protein structure prediction, the sequence track of HD-Prot is initialized according to the given protein sequence, and the structure track is completely filled with mask tokens. No matter what the length of the given protein sequence (denoted as $L$) is, HD-Prot predicts all structure tokens in one step, i.e., setting $T = 1$. By default, the structure sampling temperature $\tau_z$ = 0.0, without employing the classifier-free guidance. Such sampling settings are quite different from those used in the unconditional co-generation task. We analyze the reasons for this setup through ablation studies, and the results are presented in the table below.
>
> | Model          | Temp. $\tau_z$ | CFG | T   | CAMEO RMSD ↓       | CAMEO TM-score ↑   | PDB Date Split RMSD ↓ | PDB Date Split TM-score ↑ |
> |----------------|-------|-----|-----|--------------------|--------------------|------------------------|----------------------------|
> | HD-Prot (155M) | 0.0   | 1.0 | 1   | **9.199** ± 6.335  | **0.720** ± 0.200  | **6.231** ± 5.395      | **0.781** ± 0.181          |
> |                | 0.0   | 1.0 | L   | 9.699 ± 6.621      | 0.713 ± 0.200      | 6.654 ± 5.685          | 0.776 ± 0.185              |
> |                | 0.1   | 1.0 | L   | 9.716 ± 6.687      | 0.713 ± 0.200      | 6.653 ± 5.696          | 0.774 ± 0.188              |
> |                | 0.1   | 2.0 | L   | 9.607 ± 6.501      | 0.711 ± 0.200      | 6.599 ± 5.663          | 0.772 ± 0.187              |
> |                | 0.35  | 1.0 | L   | 9.734 ± 6.640      | 0.711 ± 0.200      | 6.648 ± 5.651          | 0.772 ± 0.187              |
> |                | 0.35  | 2.0 | L   | 9.637 ± 6.448      | 0.709 ± 0.199      | 6.592 ± 5.581          | 0.770 ± 0.188              |
> | HD-Prot (670M) | 0.0   | 1.0 | 1   | **7.468** ± 6.004  | 0.769 ± 0.177      | **5.001** ± 4.565      | 0.827 ± 0.153              |
> |                | 0.0   | 1.0 | L   | 7.500 ± 5.982      | 0.776 ± 0.177      | 5.023 ± 4.780          | **0.832** ± 0.149          |
> |                | 0.1   | 1.0 | L   | 7.525 ± 6.136      | **0.776** ± 0.176  | 5.014 ± 4.773          | 0.832 ± 0.150              |
> |                | 0.1   | 2.0 | L   | 7.743 ± 6.181      | 0.767 ± 0.177      | 5.084 ± 4.711          | 0.825 ± 0.150              |
> |                | 0.55  | 1.0 | L   | 7.757 ± 6.167      | 0.766 ± 0.177      | 5.065 ± 4.670          | 0.826 ± 0.150              |
> |                | 0.55  | 2.0 | L   | 7.848 ± 6.451      | 0.759 ± 0.178      | 5.131 ± 4.700          | 0.820 ± 0.152              |
>
> This table illustrates a clear performance hierarchy (ordered from best to worst):
> 1. "ultralow temperature ($\tau_z=0.0$) + no CFG + single pass" [It's the default setting for structure prediction];
> 2. "ultralow temperature ($\tau_z=0.0$) + no CFG + multistep iterations";
> 3. "low temperature ($\tau_z=0.1$) + w/ or w/o CFG + multistep iterations";
> 4. "higher temperature ($\tau_z=0.35/0.55$) + w/ or w/o CFG + multistep iterations" [The latter is the default setting for unconditional co-generation].
>
> We believe that the optimal settings for sampling temperature, number of iterative steps, and the use of classifier-free guidance (CFG) are closely tied to the scale and nature of the solution space associated with each task. Unconditional co-generation encourages the model to explore diverse samples across a vast joint sequence–structure distribution. Consequently, it benefits from a higher temperature, longer inference trajectories (more iterative generation steps), and greater CFG to align the structure modality with the sequence modality. In contrast, protein structure prediction is typically viewed as a near one-to-one mapping task. From the perspective of conditional generation, the input sequence serves as a highly constraining condition, leaving the model with a very limited solution space. Introducing higher sampling temperatures, more diffusion steps, or additional guidance injects unnecessary randomness into the process, which degrades prediction accuracy rather than improving it.

---

> ### Author Response · Authors · 2025-12-03
>
> ### Inverse Folding
>
> Referring to the experimental setup in DPLM-2 series [1] [2], the CAMEO 2022 and PDB Date Split datasets are used for evaluation. Compared to the nearly one-to-one structure prediction, the inverse folding has a one-to-many nature. There could be multiple distinct amino acid sequences that can fold into a target structure, in addition to its natural sequence. Therefore, rather than calculating the recovery rate of the natural sequence, our evaluation estimates the self-consistency between the target structure and refolded structure of the designed sequence. We calculate the scRMSD and scTM with the assistance of ESMFold.
>
> | Model               | CAMEO   |     | PDB Date Split | PDB Date Split |
> |---------------------|------------------|------------------|--------------------------|------------------------|
> |  | scRMSD ↓ | scTM ↑ | scRMSD ↓ | scTM ↑ |
> | MultiFlow         | –                | 0.870 ± 0.940    | –                        | 0.940 ± 0.960          |
> | ESM3                | 3.944 ± 4.964    | 0.901 ± 0.141    | 2.262 ± 3.090            | 0.940 ± 0.093          |
> | *DPLM-2 (150M)*   | 5.999 ± 7.469    | 0.848 ± 0.175    | 4.002 ± 4.700            | 0.895 ± 0.126          |
> | DPLM-2 (650M)       | 4.659 ± 4.875    | 0.871 ± 0.154    | 3.114 ± 4.034            | 0.911 ± 0.113          |
> | DPLM-2.1            | 4.304 ± 4.586    | 0.876 ± 0.141    | 2.271 ± 3.606            | 0.927 ± 0.112          |
> | *HD-Prot (155M)*  | 4.637 ± 4.730    | 0.863 ± 0.156    | 2.903 ± 3.683            | 0.919 ± 0.107          |
> | HD-Prot (670M)      | 4.675 ± 4.930    | 0.866 ± 0.151    | 2.871 ± 3.599            | 0.920 ± 0.103          |
>
> This table summarizes the performance of HD-Prot and four multimodal protein generative models, with all sampling procedures run five times with different seeds. The performance of each model on the inverse folding task is very close to its performance on the protein structure prediction task described above. ESM3 stands out the best among all methods, excels in completing the remaining multimodal context when sufficient initial information is provided. Then, HD-Prot performs highly comparable to the DPLM-2 series at both ~150M and ~650M scales. Such completely **zero-shot inverse folding results demonstrate that HD-Prot has estimated the sequence-structure joint-distribution sufficiently well**.
>
> For inverse folding, a given protein structure with length $L$ is firstly processed into the continuous structure tokens by our protein structure tokenizer. Then, the structure track of HD-Prot is initialized by those structure tokens, and the sequence track is set as fully masked. By default, HD-Prot gradually predicts all sequence tokens over $T = L$ steps with sequence sampling temperature $\tau_s = 0.1$. This sampling strategy differs from the default one used in the structure prediction task described above, and we further validate its rationale through ablation studies.
>
> | Model           | Strategy | Temp. $\tau_s$ | CAMEO scRMSD ↓ | CAMEO scTM ↑ | PDB Date Split scRMSD ↓ | PDB Date Split scTM ↑ |
> |-----------------|----------|-------|--------------|--------------|------------------------|----------------------|
> | HD-Prot (155M) | Vanilla  | 0.1   | 4.637 ± 4.730 | 0.863 ± 0.156 | 2.903 ± 3.683          | 0.919 ± 0.107        |
> |  | Argmax   | –     | 4.830 ± 4.935 | 0.861 ± 0.151 | 2.872 ± 3.511          | 0.919 ± 0.104        |
> |  | Vanilla  | 1.0   | 4.689 ± 4.812 | 0.862 ± 0.150 | 2.928 ± 3.694          | 0.919 ± 0.106        |
> | HD-Prot (670M)  | Vanilla  | 0.1   | 4.675 ± 4.930 | 0.866 ± 0.151 | 2.871 ± 3.599          | 0.920 ± 0.103        |
> |   | Argmax   | –     | 4.708 ± 4.930 | 0.864 ± 0.146 | 2.900 ± 3.591          | 0.920 ± 0.103        |
> |   | Vanilla  | 1.0   | 4.750 ± 5.350 | 0.861 ± 0.152 | 2.944 ± 3.645          | 0.918 ± 0.103        |
>
> This table shows that, with the number of sampling steps fixed at
> $T=L$, the decoding strategy for each sequence token follows a general trend (ordered from best to worst):
> 1. "low-temperature categorical sampling ($\tau_s=0.1$)" [The default setting for inverse folding];
> 2. "deterministic argmax decoding";
> 3. "higher-temperature categorical sampling ($\tau_s=1.0$)" [The default setting for unconditional co-generation].
>
> We believe this is closely related to the nature of the inverse folding task. Inverse folding is typically considered a one-to-many prediction task. It requires the generated sequence to conform to the given structural condition while still allowing for diverse valid alternatives. In other words, a small amount of randomness is better than both excessive randomness and no randomness at all.

---

> ### Author Response · Authors · 2025-12-03
>
> > W2: Small Sample sizes and Lack of Statistical Rigor
>
> We sincerely thank the reviewer for raising many important points. We would like to clarify that our evaluation protocol follows established practices in prior work. Specifically, for unconditional sequence-structure co-generation, we generate 100 samples each at protein lengths of 100, 200, 300, 400, and 500, consistent with protocols in [1] [4]. For motif-scaffolding, we generate 100 samples per subtask, as done in [1] [5]. These settings have been used in peer-reviewed publications and are widely accepted in the community for evaluating generative protein models.
>
> Meanwhile, we fully agree that generative modeling involves stochastic sampling, and results based on a single run can be sensitive to randomness. To address this concern, **we have now repeated all major experiments across five different random seeds**. In the revised manuscript, we report results as either "mean ± standard deviation" or "mean (min, max)", as appropriate, to better reflect the variability and reliability of our findings. **These updated results are included in Table 2, Table 4, and Table 9 in the revised PDF, and they consistently support our original conclusions while providing a more statistically grounded comparison**. Finally, because our claim is that HD-Prot achieves comparable, not superior, performance relative to existing methods, we do not perform statistical significance testing. Our focus is on demonstrating feasibility and competitive performance under limited compute, not on establishing statistically significant gains.

---

> ### Author Response · Authors · 2025-12-03
>
> > W5: Inappropriate Metrics
>
> Following up on the response to W2, we sincerely thank you for your rigorous and constructive feedback on our evaluation protocol. Your comments have helped us significantly improve the robustness and clarity of our experiments. Below, we address each concern in turn.
>
> ### Cluster@50 vs. #Cluster@95
> In the existing literature, #Cluster is almost universally computed with a TM-score threshold of 0.5 [1] [4] [6]. Nevertheless, we fully agree with your point: while #Cluster@50 captures the number of highly distinct structures, #Cluster@95 is needed to measure how many samples are slightly different yet non-identical. It's a finer-grained signal that helps reveal subtle mode collapse. In direct response to your suggestion, **we now report both #Cluster@50 and #Cluster@95 as complementary diversity metrics (see Table 3)**, offering a more complete picture of structural diversity across scales.
>
> ### Inner-TM
> Inner-TM (average pairwise TM-score within a set) has been widely used in prior protein generative modeling work [1] [6] to assess intra-sample diversity. While it is not meaningless, we recognize its limitations: it can be misleading when comparing models with different structural distributions. For example, MultiFlow yields pairwise TM-scores tightly clustered around 0.3–0.5, resulting in an average Inner-TM ≈ 0.35 that underrepresents its actual diversity (as confirmed by its high #Cluster@50 ≈ 0.55). Given this discrepancy and your valid concern, **we now place greater trust in the #Cluster@50 and #Cluster@95 metrics, which directly count distinct structural modes. As you suggested, we have therefore removed Inner-TM from the main results table**.
>
> ### Novelty (pdb-TM / sp-TM)
> First, it is standard practice to assess novelty by comparing generated proteins against representative databases, particularly the PDB [1] [3] [4] [6]. Omitting this evaluation would make our work inconsistent with prior studies. Meanwhile, we fully acknowledge your point: pdb-TM and sp-TM do not provide a fair assessment of memorization across all models, precisely because different methods are trained on different data. Let's recap the calculation of the pdb-TM and sp-TM, and gradually analyze their characteristics.
> - Regarding the goal of sequence-structure co-generation, for each generated protein, we use Foldseek with 3Di + amino acid Gotoh-Smith-Waterman alignment to search against the full PDB (for pdb-TM) or Swiss-Prot (for sp-TM) database, and report the maximum TM-score against any entry. That is, the similarity to the most similar known protein in terms of both sequence and structure. Please see Appendix D.4 for the detailed Foldseek command.
> - For HD-Prot and DPLM-2, whose training data (both sequences and structures) are strict subsets of PDB and Swiss-Prot, this pipeline reliably detects true memorization: if a model regurgitates a training example, it will match a database entry and yield a high TM-score.
> - However, for MultiFlow, ESM3, and La-Proteina, the situation is different. These models incorporate external data beyond PDB/SwissProt, such as synthetic sequences from inverse folding models, or structure data from AFDB that extends beyond Swiss-Prot. Consequently, they may generate proteins that are memorized from their own extended training sets, yet absent from PDB/SwissProt. In such cases, pdb-TM/sp-TM will incorrectly label these as "novel", making their novelty scores overly optimistic.
>
> Unfortunately, we cannot fairly recompute novelty against each model's private training corpus. Downloading each model’s training data and building dedicated similarity search tools for them would be far too labor-intensive and cumbersome. Therefore, **we suggest interpreting pdb-TM/sp-TM not as a direct measure of memorization, but as a measure of dissimilarity to the canonical PDB and SwissProt universe**. For some models, high novelty may reflect genuine generalization; for others, it may stem from training on expanded data—a valid engineering strategy, and indeed a credit to those teams. But the metric itself cannot distinguish between the two.

---

> ### Author Response · Authors · 2025-12-03
>
> ### scRMSD
> Our use of averaged scRMSD strictly follows the evaluation protocol of DPLM-2 [1], our primary baseline, which ensures a fair comparison. Meanwhile, **we also provide the unaveraged, per-length results for scRMSD and other important metrics in Appendix Figure 3. These detailed results confirm that our main conclusions do not change**: HD-Prot and the DPLM-2 series achieve comparable strong performance.
>
> ### Training data discrepancies
> As a matter of fact, models use different training data and scales, which have a significant impact on the performance of the models. **We have now added a summary of the training data scales for all compared methods in Table 2**. It shows that HD-Prot is trained on a considerably smaller dataset than ESM3 and La-Proteina. Nevertheless, it already achieves strong performance. This highlights the learning efficiency of our approach and suggests clear potential for further improvement through scaling up.
>
> We hope these revisions and clarifications address your concerns. Thank you again for pushing us to strengthen our evaluation. Please visit Table 2 of the revised PDF for more details.

---

> ### Author Response · Authors · 2025-12-03
>
> > W3: Missing Baselines
>
> Thank you for the suggestion. We would like to clarify the scope of our work.
>
> Protein generative modeling is a broad field with multiple development directions. One well-established approach uses a pipeline of separate models: generate a structure, then design a sequence with an inverse folding model like ProteinMPNN, and finally validate with a structure predictor like ESMFold. Another, **more recent direction aims to build unified multimodal models that directly learn the joint distribution of sequences and structures in a single system. Our paper explicitly belongs to this second direction**.
>
> We do not aim to justify the existence of multimodal modeling that has been motivated by multiple recent works [1] [2] [3] [4] [7] [13]. Nor do we claim to prove its long-term superiority, that will require broader validation over time. Instead, our goal is to advance within the multimodal modeling community, showing that a sequence-based protein language model can be extended to model continuous structural information while preserving its sequence knowledge. Because of this focus, we compare HD-Prot against other multimodal generative models, not those unimodal methods. Comparing against a full “structure → ProteinMPNN → ESMFold” pipeline would mix two different modeling philosophies and **go beyond our scope**.
>
> Meanwhile, we note that calling pipeline methods "simple" and unified models "complex" may reflect familiarity rather than actual complexity. Pipeline components like RFDiffusion, ProteinMPNN or ESMFold appear simple because they are well-packaged and widely used, but the full workflow involves multiple stages, each with its own assumptions and operational details. In contrast, unified models are still new, so their learning and inference processes are more visibly reviewed. The long-term goal of the multimodal direction is exactly to make end-to-end protein design and cross-modal predictions as easy as calling a single model. When that happens, it will also seem simple.
>
> Finally, we are using the salad autoencoder as structure tokenizer. It only reconstructs given structures and cannot sample new ones. Therefore, it is not a valid baseline for generation tasks.

---

> ### Author Response · Authors · 2025-12-03
>
> > W4: Missing Hyperparameter Analysis
>
> We appreciate the reviewer’s careful reading, but respectfully disagree with the characterization of our approach as “limited novelty.” While HD-Prot does build upon diffusion-based generative modeling, its core contribution is conceptual: it challenges a popular assumption that protein structures should be discretized to be integrated into protein language models [1] [7] [8]. Our work demonstrates that a protein language model can jointly model discrete sequences and continuous structures within a unified framework. It has the potential to leverage pre-trained sequence knowledge while estimating high-fidelity structural information simultaneously and seamlessly. As you acknowledged in Strongth 5, this redefines the design space for multimodal pLMs and opens a new direction, which we believe constitutes significant novelty.
>
> That said, we fully agree that deeper ablation studies strengthen scientific rigor. Below, we clarify our hyperparameter choices, separating training-time and sampling-time settings.
>
> 1. Training hyperparameters
>
> As detailed in Appendix D.2, our training setup follows established practices:
> - For the sequence track, we adopt the reweighting scheme from [9], concretely formulated as $\lambda^{(t_s)} = 1 - (t_s - 1) / T$.
> - For the structure track, we use a constant weight $\lambda^{(t_z)} = 1$, consistent with [10].
> - The diffusion schedule $\beta_{t'}$ is a standard linear schedule.
> - The loss balancing coefficient $\gamma$ is set to 0.2 empirically, chosen to align the magnitudes of sequence and structure losses.
>
> We acknowledge that more extensive tuning (e.g., search over non-linear $\lambda^{(t_s)}$, non-constant $\lambda^{(t_z)}$, and non-linear $\beta_{t'}$ schedules) could yield marginal gains. However, our goal is to **demonstrate the feasibility of continuous-token integration under realistic resource limits**, not to exhaustively optimize engineering. Future work with larger budgets can certainly refine these choices.
>
> 2. Sampling hyperparameters
>
> In contrast, **we have conducted comprehensive ablations on sampling strategies across all four tasks (unconditional co-generation, motif scaffolding, structure prediction, and inverse folding), as reported in Appendix E.5**.
>
> Specific to unconditional sequence-structure co-generation, we conducted a grid search over the structure sampling temperature $\tau_z$ and CFG scale for both HD-Prot (155M) and HD-Prot (670M). The results are visualized as **heatmaps in Figures 8 and 9 of the revised manuscript**. Our findings align with established trends in generative modeling: higher sampling temperatures increase structural diversity but may compromise sample quality. Regarding CFG, moderate guidance (scale = 2.0 or 2.5) consistently yields better pLDDT, scRMSD, and scTM scores than either no guidance (scale = 1.0) or overly strong guidance (scale = 3.0). At the same time, larger CFG scales generally enhance diversity.
>
> When comparing among the four tasks, our key insight is: **Optimal sampling depends on the task’s condition strength and solution space size**. For low-condition task with large solution space (e.g., unconditional co-generation), higher temperature (e.g., $\tau_z = 0.35/0.55$), more steps ($T = L$), and classifier-free guidance (CFG) improve diversity and sequence-structure self-consistency. For high-condition task with very limited solution space (e.g., structure prediction), low temperature ($\tau_z = 0.0$), single-step sampling, and no CFG yield the best accuracy.
>
> We hope these clarifications and new analyses demonstrate both the novelty of our framework and our commitment to empirical thoroughness.

---

> ### Author Response · Authors · 2025-12-03
>
> > W6 & W12: Inappropriate wording
>
> We appreciate the reviewer’s attention to wording precision. Regarding Weakness 6, the phrase “highly competitive performance” appeared only once in the abstract, and was intended to convey that our method achieves performance on par with existing state-of-the-art models, not superior to them. To avoid any potential overstatement, we have revised it to simply “competitive performance.”
>
> Regarding Weakness 12, we agree that our original wording about generalization to longer proteins could be misinterpreted. The original sentence states that our model “possesses the potential to be generalized to generate larger proteins”, itself a gentle claim aimed at qualitative analysis. Now, in order to avoid any misunderstandings, we have moved all qualitative co-generation examples, including the 600-700 residue cases, to Appendix E.4. Meanwhile, we revised the related text to state only what the evidence supports: “Although our model was trained primarily on proteins shorter than 512 residues, we can still find certain good cases for larger proteins with 600-700 residues.” We no longer make any claim about generalization capability beyond observation of plausible samples.
>
> Thank you for prompting us to improve the clarity and rigor of our language.

---

> ### Author Response · Authors · 2025-12-03
>
> > W7 & Q1: Quantification of Information Loss
>
> We thank the reviewer for raising the important clarification about information loss. We would like to emphasize that we never claim that continuous representations are lossless in our paper. **Our argument is not that continuous structure tokens eliminate information loss entirely, but rather that they reduce it compared to discretization-based approaches**. Specifically, we state that: "the quantization process inevitably compresses and omits portions of continuous information, leading to the loss of fine-grained structural details and imprecise geometric relationships." In contrast, the continuous structure tokens are described as: "high-fidelity protein structure latents that avoid vector quantization," which "enhance continuous information fidelity" and "benefit generative modeling with greater expressive capability in representing fine-grained structural knowledge."
>
> We fully agree with the reviewer that any mapping from raw coordinates to a latent space is inherently lossy. However, the key distinction lies in the degree of information preservation. Discretization (e.g., via VQ-VAE) introduces an additional, non-differentiable, and relatively coarse bottleneck that further discards continuous geometric details. By avoiding quantization, continuous tokens retain more of the original structural signal, thereby raising the upper bound of what a generative pLM can achieve. Regarding the latent dimension, we apologize for omitting this detail in the main text previously. $d_{struct}=20$, and we have now added this specification in Section 3.1. Notably, the mapping from $L × 4 × 3$ to $L × 20$ is not a “dimensionality reduction” in the conventional sense.
>
> Furthermore, we would like to clarify that **a quantitative comparison of reconstruction fidelity is already provided in the paper (Section 3.1 and Table 1)**.
> | Tokenizer   | CAMEO scRMSD ↓ | CAMEO scTM ↑ |
> |-------------|----------------|--------------|
> | DPLM-2      | 2.109 ± 2.635  | 0.940 ± 0.080|
> | ESM3        | 1.225 ± 2.485  | 0.975 ± 0.072|
> | salad-vq    | 0.842 ± 0.913  | 0.981 ± 0.042|
> | **salad**   | 0.393 ± 1.072  | 0.996 ± 0.015|
>
> Specifically, we evaluate the structure reconstruction quality of the continuous tokenizer we used (named "salad") against the discrete tokenizers of DPLM-2 and ESM3, and a VQ-VAE variant of salad itself. It is observed that the continuous tokenizer achieves obviously better reconstruction accuracy, demonstrating that avoiding vector quantization preserves more fine-grained structural information.

---

> ### Author Response · Authors · 2025-12-03
>
> > W8, Q3 & Q4: Motif-Scaffolding Evaluation and Implementation
>
> We appreciate the reviewer’s insightful comments regarding motif-scaffolding evaluation and implementation. Below we address each point in turn.
>
> 1. The choice of benchmark
>
> We acknowledge that MotifBench [11] is now the better benchmark for motif-scaffolding, as the RFDiffusion benchmark has largely been saturated by recent single-modality structure generation models. However, the development of multimodal protein models for motif-scaffolding tasks remains relatively underdeveloped, and the RFDiffusion benchmark is still widely recognized. To the best of our knowledge, existing multimodal pLMs [1] [7] and protein co-design methods [4] have so far been evaluated exclusively on the RFDiffusion motif-scaffolding benchmark. While we fully agree that migrating the community of multimodal protein models to MotifBench is a valuable future direction, such an effort would require a substantial undertaking beyond the scope of this work.
>
> Meanwhile, our results (Table 4 and Table 9 in the manuscript) demonstrate that ESM3, DPLM-2, and HD-Prot successfully solve approximately 14 to 21 out of 24 problems. It suggests that there is considerable room for improvement, and meaningful distinctions in model capabilities remain observable within this context. Our claim is modest: HD-Prot can perform motif-scaffolding at a level comparable to existing multimodal pLMs. For this purpose, the RFDiffusion benchmark remains appropriate and sufficient for the evaluation.
>
> 2. How motif tokens are "maintained unchanged"
>
> **The motif-scaffolding procedure of HD-Prot is detailed in Appendix C.2** in the revised manuscript: "Given a motif of length $l$, its sequence is mapped to discrete sequence tokens, and its structure is encoded into continuous structure tokens via our tokenizer. Based on the motif’s position and total scaffold length $L$, the input consists of fixed motif tokens at their designated positions, with all other positions masked. HD-Prot then performs generation over $T=L−l$ steps, updating only the masked positions while keeping the initialized motif tokens fixed throughout. After generation, all tokens are decoded back to sequences and 3D coordinates." Thus, both sequence and structure tracks of the motif are preserved unchanged during sampling.
>
> Besides, we have performed an ablation study that compares two strategies: (a) sampling all $L$ positions (allowing motif tokens to be overwritten) vs. (b) sampling only $L−l$ masked positions (keeping motif fixed). **Results in Appendix E.5.2 show that strategy (b) yields slightly better success rates**, confirming that preserving the motif is beneficial, though the difference is modest.
>
> 3. Why classifier-free guidance (CFG) is not used in motif-scaffolding
>
> Thank you for your valuable questions, we have conducted a series of ablation studies discussing the effect of the key sampling strategies across different tasks, see Appendix E.5. Figures 8 and 9 in Appendix E.5.1 demonstrate that CFG enhances self-consistency and diversity metrics in unconditional co-generation. However, Table 10 in Appendix E.5.2 indicates that applying CFG in motif-scaffolding usually leads to performance degradation.
>
> We believe this is because **our current CFG implementation is misaligned with the motif-scaffolding objective**. As described in Appendix C.3, CFG steers structure tokens to be more consistent with the generated sequence at each step. In motif-scaffolding, however, both sequence and structure must align with the fixed input motif, not with each other. Introducing CFG thus creates conflicting optimization signals, pulling the scaffold away from the motif constraint.

---

> ### Author Response · Authors · 2025-12-03
>
> > W9 & Q5: Scalability & Modality Expansion
>
> We sincerely appreciate the reviewer’s thoughtful critique. Below we clarify two central points: the scalability of our approach, and our contribution as modality expansion.
>
> 1. Scalability
>
> While we acknowledge that our current implementation is not sufficiently scaled up, **we emphasize that this reflects an engineering constraint, not a fundamental limitation of our approach**. Our work operates within the well-established scaling laws: model performance jointly depends on model size, training data volume, and compute budget [12]. Unfortunately, due to resource limitations, the training of HD-Prot was conducted on a single GPU with only ~210K training examples, and we could not explore further large-scale settings. The failures noted, e.g., training from scratch or full fine-tuning of the 650M model, are entirely expected under such practical constraints. Modern protein language models are trained on tens of millions of sequences [1] [13]. Attempting to train or fully fine-tune pLMs on two orders of magnitude less data would naturally lead to suboptimal results or collapse, regardless of the modeling framework. **This reflects data/compute scarcity, not a flaw in our architecture**.
>
> Critically, our approach is not fundamentally constrained to small models or limited data. The hybrid diffusion architecture, which embeds a continuous diffusion head atop a discrete pLM, is modular and compatible with any scale of backbone. With sufficient data and compute, we fully expect our HD-Prot model to scale gracefully. In fact, the very fact that **we achieve competitive performance despite severe resource limitations underscores the efficiency and promise of our design**.
>
> 2. Modality Expansion
>
> In both W9 and Q5, the reviewer appears to equate "using a pre-trained model" with "lack of contribution." We respectfully disagree. A major trend in modern AI is post-training adaptation, including modality expansion. It is a valuable contribution for well-resourced teams to develop pre-trained foundation models that encode rich knowledge. Equally valuable are efforts by other teams to extend these models, for example, by injecting multimodal knowledge or adapting them to new applications.
>
> From ESM-2 to DPLM, a pLM that originally possessed only representation capability was extended to support both representation and generation. Then, our contribution lies in demonstrating that a sequence-only pLM (DPLM) can be seamlessly extended to jointly model continuous structural information without discretization, while preserving its sequence modeling capability. HD-Prot shows that discrete and continuous tokens can coexist in a unified generative framework. We believe this represents a meaningful methodological advance for multimodal protein language modeling.

---

> ### Author Response · Authors · 2025-12-03
>
> > W10: Hyperparamter-Sensitive Sampling
>
> Thank you for highlighting this point. We sincerely appreciate the opportunity to provide further clarification. We would like to address your concerns regarding the characterization of our method as "brittle" or "hyperparameter-sensitive" from two perspectives.
>
> First, the choice between full fine-tuning and LoRA is driven by practical considerations of compute, data scale, and parameter efficiency, not fundamental instability. Both strategies aim to control the effective number of trainable parameters, which is a common practice when adapting large models under resource constraints. This is a matter of engineering trade-off, not a sign of brittleness.
>
> Second, the need to tune sampling temperature and classifier-free guidance is universal across generative models, not unique to HD-Prot. In protein generative models, as well as image generation models (e.g., Stable Diffusion), the temperature and CFG are standard knobs that significantly affect output diversity and quality [10]. It is entirely expected that different model scales or training configurations yield different optimal temperatures. This reflects differences in the learned data distributions, not any inherent instability of the models.
>
> Besides, please see Appendix E.5 of the revised manuscript for a detailed analysis of HD-Prot’s sampling hyperparameters.

---

> ### Author Response · Authors · 2025-12-03
>
> > W11: No failure mode analysis
>
> We thank the reviewer for this valuable suggestion. In response, **we have added a dedicated failure mode analysis in Appendix E.4, which includes a detailed case study of a representative poor-generation example (visualized in Figure 7.B)**.
>
> In this case, ESMFold assigns a globally high average pLDDT (74.609), yet a short coil segment connecting an alpha-helix to the rest of the structure shows notably lower per-residue pLDDT (between 50 and 70), indicating uncertainty in the helix orientation. Structural alignment further reveals that the co-generated and ESMFold-predicted structures place this helix in markedly different directions, leading to a very poor RMSD score. Moreover, the co-generated helix itself exhibits non-physical distortions, suggesting that the underlying continuous structure tokens remain noisy.
>
> From this representative example, we identify two common error patterns:
> - Misjudged structural orientation when the model encounters either unreasonable sequence fragments or biologically authentic disordered regions;
> - Local structural collapse when the quality of the corresponding continuous tokens is low.
>
> Regarding length-dependent performance degradation (Figure 6), we state that it stems from “less training data at longer lengths”. This statement is indeed brief, but it could reflect an empirical fact. All natural proteins, regardless of length, obey fundamental physical and evolutionary constraints. However, those constraints have not been explicitly elucidated, and current AI models learn these principles implicitly through data. Since longer proteins are inherently underrepresented in structural databases such as PDB and AFDB, generative protein models almost inevitably receives fewer examples to estimate their distributions, leading to reduced performance.

---

> ### Author Response · Authors · 2025-12-03
>
> > Q2: Impacts of Tokenization Quality
>
> We thank the reviewer for raising this question about the dependence of HD-Prot on the quality of the continuous structure tokenizer.
>
> First, we clarify that "requiring scRMSD < 1.0 and scTM > 0.9" is a data filtering criterion, analogous to filtering by sequence length < 512 or AFDB pLDDT scores > 85. Its purpose is to ensure that the model learns from high-fidelity structural representations during training. Just as training on low-confidence or misfolded structures would introduce noise, learning from poorly reconstructed continuous tokens would mislead the diffusion process, as the target distribution would not faithfully reflect the true 3D geometry. We expect that including poorly reconstructed samples would introduce noise into the training targets and likely degrade performance. Second, we agree that HD-Prot’s performance is ultimately limited by the quality of the continuous structure tokenizer, and this limitation applies to all multimodal pLMs that rely on structural tokenization.

---

> ### Author Response · Authors · 2025-12-03
> **References**
>
> [1] Wang et al. DPLM-2: A Multimodal Diffusion Protein Language Model. ICLR, 2025.
> [2] Hsieh et al. Elucidating the Design Space of Multimodal Protein Language Models. ICML, 2026.
> [3] Campbell et al. Generative Flows on Discrete State-Spaces: Enabling Multimodal Flows with Applications to Protein Co-Design. ICML, 2024.
> [4] Geffner et al. La-Proteina: Atomistic Protein Generation via Partially Latent Flow Matching. Arxiv, 2025.
> [5] Yim et al. Improved motif-scaffolding with SE(3) flow matching. TMLR, 2024.
> [6] Geffner et al. Proteina: Scaling Flow-based Protein Structure Generative Models. ICLR, 2025.
> [7] Hyas et al. Simulating 500 million years of evolution with a language model. Science, 2025.
> [8] Yuan et al. Protein Structure Tokenization: Benchmarking and New Recipe. ICML, 2025.
> [9] Wang et al. Diffusion Language Models Are Versatile Protein Learners. ICML, 2024.
> [10] Li et al. Autoregressive Image Generation without Vector Quantization. NeurIPS, 2024.
> [11] Zheng et al. MotifBench: A standardized protein design benchmark for motif-scaffolding problems. Arxiv, 2025.
> [12] Kaplan et al. Scaling Laws for Neural Language Models. Arxiv, 2020.
> [13] Su et al. SaProt: Protein Language Modeling with Structure-aware Vocabulary. ICLR, 2024.

---

### Official Review · Reviewer_tjPD · 2025-11-01

**Soundness:** 3
**Presentation:** 3
**Contribution:** 3
**Rating:** 4
**Confidence:** 3

**Summary:**

This paper presents HD-Prot, a hybrid diffusion protein language model that jointly generates protein sequences and structures using continuous structure tokens instead of quantized ones. By integrating a non-quantized autoencoder (salad) and a unified diffusion framework operating over both discrete (sequence) and continuous (structure) modalities, HD-Prot avoids information loss from discretization. Experiments show that it achieves competitive sequence–structure co-generation and motif-scaffolding performance relative to state-of-the-art multimodal pLMs (e.g., DPLM-2, ESM3) while requiring far fewer computational resources.

**Strengths:**

1. **Continuous structure tokens** – High-fidelity structure representation without quantization loss (scRMSD ≈ 0.39 Å).
2. **Unified diffusion modeling** – Seamlessly bridges discrete and continuous modalities within one LM.
3. **Compute-efficient design** – Matches large multimodal pLMs’ performance using only a single-GPU setup.

**Weaknesses:**

1. The model does not outperform existing co-design approaches on most benchmarks presented, suggesting there is room for further improvement.

**Questions:**

1. Did the authors train the protein structure autoencoder themselves, or was it directly adopted from prior work? If trained in-house, was it trained on the same **PDB + AFDB-SwissProt** dataset as used for HD-Prot?
2. Can the model also perform **structure prediction** and **inverse folding** tasks? If so, it would strengthen the work to include analyses comparable to those presented in **DPLM-2**.

---

> ### Author Response · Authors · 2025-12-03
>
> We sincerely thank you for the thoughtful feedback and for recognizing the key contributions of our work, i.e., use of continuous structure tokens for high-fidelity representation, the unified diffusion framework that bridges modalities, and the competitive performance achieved under limited computational resources. We address the specific questions and suggestions below to further strengthen the manuscript.
>
> > W1: Competitive but not SOTA performance
>
> We thank the reviewer for this fair observation. We readily acknowledge that HD-Prot does not surpass all existing co-design methods on every benchmark. However, **this modest performance stems not from a flaw in the method itself, but from severe practical constraints**. Due to limited computational resources, we adopted several engineering compromises: pre-cached tokenization, mixed-precision training, small batch sizes, and training on just a single GPU. As detailed in Appendix E.2, our total training cost is less than one twentieth of that of DPLM-2 when measured in cloud rental price (device × days × rate).
>
> Despite these limitations, HD-Prot achieves highly competitive performance with DPLM-2 across multiple tasks, including unconditional co-generation, motif scaffolding, protein structure prediction, and inverse folding. This demonstrates that continuous structure tokens are not only viable but also computationally efficient, offering a new pathway for multimodal protein modeling that avoids the information loss inherent in discretization.
>
> **Our goal was not to achieve state-of-the-art performance at all costs, but to demonstrate the feasibility of integrating continuous structure tokens into protein language models**. The fact that HD-Prot achieves competitive results under such limitations already validates the feasibility of our approach, and we believe this constitutes a meaningful contribution to the field.
>
> > Q1: Tokenizer Implementation
>
> Thank you for your question. We did not train the protein structure autoencoder ourselves. We adopted the pre-trained Salad autoencoder from prior work [1]. Its training data consists of PDB entries submitted before 31 December 2020. It is partially overlapped with the PDB data we used, while it does not involve AFDB-SwissProt at all.
>
> [1] Jendrusch et al. Efficient protein structure generation with sparse denoising models. Nature Machine Intelligence, 2025.

---

> ### Author Response · Authors · 2025-12-03
>
> > Q2: Missing Evaluation Tasks
>
> We sincerely thank the reviewer for this constructive suggestion. We agree that a comprehensive evaluation of a multimodal protein language model should go beyond unconditional co-generation and motif scaffolding. As you suggested, **we have added two new evaluation tasks in the revised manuscript: Protein Structure Prediction, and Inverse Folding**. We have carefully revised the manuscript to incorporate the two newly added tasks. Please refer to the text highlighted in blue in our updated PDF. Specifically:
> - We made minor adjustments to the wording in the **Abstract, Introduction, Preliminaries, and Conclusion** sections to better reflect the expanded scope.
> - In **Sections 4.4 and 4.5**, we present the evaluation setups and results for the protein structure prediction and inverse folding.
> - In **Appendix C.2**, we provide a detailed description of HD-Prot's inference procedures for these two tasks.
> - In **Appendix D.3**, we detail how the baseline methods are implemented for these tasks.
> - In **Appendix E.5.3 and E.5.4**, we provide ablation studies on the sampling hyperparameters for these tasks.
>
> Here, we focus on discussing the main experimental results.
>
> ### Protein Structure Prediction
>
> Following the experimental setup of DPLM-2 series, we evaluate the protein structure prediction capability of multimodal protein generative models via two datasets, i.e., CAMEO 2022, and a PDB Date Split curated by MultiFlow. The structure prediction results are compared to the corresponding native structures, and the RMSD and TM-score are calculated to assess the prediction accuracy.
>
> | Model                | CAMEO |       | PDB Date Split | |
> |----------------------|------------------|----------------|-------------------|----------------|
> |                      | RMSD ↓           | TM-score ↑     | RMSD ↓            | TM-score ↑     |
> | MultiFlow            | 17.840 ± 17.96   | 0.810 ± 0.880  | 15.640 ± 16.08    | 0.530 ± 0.490  |
> | ESM3         | 5.377 ± 6.303    | 0.860 ± 0.168  | 4.042 ± 4.824     | 0.883 ± 0.150  |
> | *DPLM-2 (150M)*    | 9.919 ± 6.994    | 0.720 ± 0.189  | 7.833 ± 6.004     | 0.765 ± 0.169  |
> | DPLM-2 (650M)        | 7.483 ± 6.126    | 0.786 ± 0.170  | 5.253 ± 5.143     | 0.836 ± 0.144  |
> | DPLM-2.1      | 6.272 ± 6.202    | 0.824 ± 0.166  | 2.869 ± 3.942     | 0.915 ± 0.113  |
> | *HD-Prot (155M)*   | 9.185 ± 6.316    | 0.719 ± 0.201  | 6.229 ± 5.391     | 0.781 ± 0.181  |
> | HD-Prot (670M)       | 7.468 ± 6.004    | 0.769 ± 0.177  | 5.001 ± 4.565     | 0.827 ± 0.153  |
>
> This table presents the statistical comparison between HD-Prot and four multimodal protein generative models, where all predictions with randomness are repeated five times with different seeds. The performance of the baseline methods on the Protein Structure Prediction aligns well with our observations in the Unconditional Sequence–Structure Co-Generation evaluation. ESM3 has undergone ultra-large-scale multimodal masked language modeling pretraining. However, its pretraining contains relatively few examples with high mask ratios, which limits its performance in unconditional co-generation. In the structure prediction task, where the full sequence is provided as input, ESM3’s conditional generation capability is fully leveraged, leading to strong performance. Meanwhile, the DPLM-2 series consistently present sufficiently good structure prediction performance, demonstrating mastery of cross-modal knowledge. Notably, HD-Prot performs better or comparable to the DPLM-2 at both ~150M and ~650M scales. During the training of HD-Prot, it has never seen a situation where the sequence track is completely given and the structure track is fully masked. This absolutely **zero-shot structure prediction performance indicates that HD-Prot has acquired considerably strong sequence-structure cross-modal capabilities**.

---

> ### Author Response · Authors · 2025-12-03
>
> ### Inverse Folding
>
> Referring to the experimental setup in [1] [2], the CAMEO 2022 and PDB Date Split datasets are used for evaluation. Compared to the nearly one-to-one structure prediction, the inverse folding has a one-to-many nature. There could be multiple distinct amino acid sequences that can fold into a target structure, in addition to its natural sequence. Therefore, rather than calculating the recovery rate of the natural sequence, our evaluation estimates the self-consistency between the target structure and refolded structure of the designed sequence. We calculate the scRMSD and scTM with the assistance of ESMFold.
>
> | Model               | CAMEO   |     | PDB Date Split | PDB Date Split |
> |---------------------|------------------|------------------|--------------------------|------------------------|
> |  | scRMSD ↓ | scTM ↑ | scRMSD ↓ | scTM ↑ |
> | MultiFlow         | –                | 0.870 ± 0.940    | –                        | 0.940 ± 0.960          |
> | ESM3                | 3.944 ± 4.964    | 0.901 ± 0.141    | 2.262 ± 3.090            | 0.940 ± 0.093          |
> | *DPLM-2 (150M)*   | 5.999 ± 7.469    | 0.848 ± 0.175    | 4.002 ± 4.700            | 0.895 ± 0.126          |
> | DPLM-2 (650M)       | 4.659 ± 4.875    | 0.871 ± 0.154    | 3.114 ± 4.034            | 0.911 ± 0.113          |
> | DPLM-2.1            | 4.304 ± 4.586    | 0.876 ± 0.141    | 2.271 ± 3.606            | 0.927 ± 0.112          |
> | *HD-Prot (155M)*  | 4.637 ± 4.730    | 0.863 ± 0.156    | 2.903 ± 3.683            | 0.919 ± 0.107          |
> | HD-Prot (670M)      | 4.675 ± 4.930    | 0.866 ± 0.151    | 2.871 ± 3.599            | 0.920 ± 0.103          |
>
> This table summarizes the performance of HD-Prot and four multimodal protein generative models, with all sampling procedures run five times with different seeds. The performance of each model on the inverse folding task is very close to its performance on the protein structure prediction task described above. ESM3 stands out the best among all methods, excels in completing the remaining multimodal context when sufficient initial information is provided. Then, HD-Prot performs highly comparable to the DPLM-2 series at both ~150M and ~650M scales. Such completely **zero-shot inverse folding results demonstrate that HD-Prot has estimated the sequence-structure joint-distribution sufficiently well**.

---

### Official Review · Reviewer_TFWS · 2025-11-02

**Soundness:** 2
**Presentation:** 3
**Contribution:** 3
**Rating:** 4
**Confidence:** 4

**Summary:**

This paper introduces HD-Prot, a hybrid diffusion protein language model designed to address a key limitation in existing multimodal protein models: the information loss caused by discretizing continuous 3D protein structures. Instead of using discrete tokens, HD-Prot represents protein structures as high-fidelity continuous structure tokens generated by a non-quantized autoencoder. The model extends a sequence-only protein language model to jointly process both discrete amino acid sequence tokens and these new continuous structure tokens. The HD-Prot leverages a unified absorbing diffusion process (masking and denoising) to capture dependencies at the inter-token level between both the sequence and structure modalities. For the sequence and structure prediction, the HD-Prot employs categorical prediction for the discrete sequence and MAR-like continuous diffusion for the structure tokens. Experiments demonstrate that HD-Prot achieves competitive performance on par with state-of-the-art multimodal protein language models in unconditional sequence-structure co-generation and motif-scaffolding tasks.

**Strengths:**

1. Proposing novel approach HD-Prot, which uses continuous structure tokens and circumvents the need for vector quantization, to avoid structural information loss for multimodal protein language models.
2. The proposed hybrid diffusion framework effectively bridges the discrete-continuous modality gap, where a unified backbone is used to process the inter-token correlation and separate categorical and denoising prediction heads for sequence and structure, respectively.
3. Despite its low training cost, HD-Prot achieves results that are competitive with or on par with state-of-the-art models in unconditional co-generation and motif-scaffolding tasks.

**Weaknesses:**

1. The experiments are limited to co-generation and motif-scaffolding, lacking the other representative conditional generation tasks such as folding and inverse folding, which are also evaluated by other multimodal pLMs (DPLM-2, ESM-3, DPLM-2.1). This leaves the model's true multimodal capabilities unverified.
2. For the unconditional co-generation experiments, the authors should provide a comparison against the recent work La-Proteina [1].
3. The authors argue that a fundamental limitation of structure quantization is information loss of fine-grained structural details, and they leverage continuous structure tokens to address this. In Table 2, HD-Prot achieves superior designability compared to ESM3 and DPLM2. However, this result cannot directly substantiate the advantage of continuous structure tokens. A critical factor is that HD-Prot, ESM3, and DPLM2 utilize different structure tokenizers. Therefore, the performance discrepancy might originate from the tokenizers themselves, rather than from the distinction between continuous and discrete representations.
4. Regarding motif-scaffolding, although the method solves a greater number of problems than other baselines, its average success rate is lower than DPLM-2. Furthermore, the success rate for several specific problems is extremely low (e.g., 5IUS: 1/100, 5WN9: 1/100, 4ZYP: 2/100). This suggests that the reported number of solved problems may be influenced by stochasticity rather than representing a robust capability.

[1] La-Proteina: Atomistic Protein Generation via Partially Latent Flow Matching

**Questions:**

See above weaknesses.

---

> ### Author Response · Authors · 2025-12-03
>
> We sincerely thank the reviewer for the insightful assessment of our work, particularly for recognizing the novelty of HD-Prot in leveraging continuous structure tokens to avoid vector quantization, the effectiveness of our hybrid diffusion framework in unifying discrete and continuous modeling, and the competitive performance achieved under low training cost. Below, we address each of your concerns in turn.
>
> > W1: Missing Evaluation Tasks
>
> We sincerely thank the reviewer for this constructive suggestion. We agree that a comprehensive evaluation of a multimodal protein language model should go beyond unconditional co-generation and motif scaffolding. As you suggested, **we have added two new evaluation tasks in the revised manuscript: Protein Structure Prediction, and Inverse Folding**. We have carefully revised the manuscript to incorporate the two newly added tasks. Please refer to the text highlighted in blue in our updated PDF. Specifically:
> - We made minor adjustments to the wording in the **Abstract, Introduction, Preliminaries, and Conclusion** sections to better reflect the expanded scope.
> - In **Sections 4.4 and 4.5**, we present the evaluation setups and results for the protein structure prediction and inverse folding.
> - In **Appendix C.2**, we provide a detailed description of HD-Prot's inference procedures for these two tasks.
> - In **Appendix D.3**, we detail how the baseline methods are implemented for these tasks.
> - In **Appendix E.5.3 and E.5.4**, we provide ablation studies on the sampling hyperparameters for these tasks.
>
> Here, we focus on discussing the main experimental results.
>
> ### Protein Structure Prediction
>
> Following the experimental setup of DPLM-2 series, we evaluate the protein structure prediction capability of multimodal protein generative models via two datasets, i.e., CAMEO 2022, and a PDB Date Split curated by MultiFlow. The structure prediction results are compared to the corresponding native structures, and the RMSD and TM-score are calculated to assess the prediction accuracy.
>
> | Model                | CAMEO |       | PDB Date Split | |
> |----------------------|------------------|----------------|-------------------|----------------|
> |                      | RMSD ↓           | TM-score ↑     | RMSD ↓            | TM-score ↑     |
> | MultiFlow            | 17.840 ± 17.96   | 0.810 ± 0.880  | 15.640 ± 16.08    | 0.530 ± 0.490  |
> | ESM3 (1.4B)          | 5.377 ± 6.303    | 0.860 ± 0.168  | 4.042 ± 4.824     | 0.883 ± 0.150  |
> | *DPLM-2 (150M)*    | 9.919 ± 6.994    | 0.720 ± 0.189  | 7.833 ± 6.004     | 0.765 ± 0.169  |
> | DPLM-2 (650M)        | 7.483 ± 6.126    | 0.786 ± 0.170  | 5.253 ± 5.143     | 0.836 ± 0.144  |
> | DPLM-2.1      | 6.272 ± 6.202    | 0.824 ± 0.166  | 2.869 ± 3.942     | 0.915 ± 0.113  |
> | *HD-Prot (155M)*   | 9.185 ± 6.316    | 0.719 ± 0.201  | 6.229 ± 5.391     | 0.781 ± 0.181  |
> | HD-Prot (670M)       | 7.468 ± 6.004    | 0.769 ± 0.177  | 5.001 ± 4.565     | 0.827 ± 0.153  |
>
> This table presents the statistical comparison between HD-Prot and four multimodal protein generative models, where all predictions with randomness are repeated five times with different seeds. The performance of the baseline methods on the Protein Structure Prediction aligns well with our observations in the Unconditional Sequence–Structure Co-Generation evaluation. ESM3 has undergone ultra-large-scale multimodal masked language modeling pretraining. However, its pretraining contains relatively few examples with high mask ratios, which limits its performance in unconditional co-generation. In the structure prediction task, where the full sequence is provided as input, ESM3’s conditional generation capability is fully leveraged, leading to strong performance. Meanwhile, the DPLM-2 series consistently presents sufficiently good structure prediction performance, demonstrating mastery of cross-modal knowledge. Notably, HD-Prot performs better or comparable to the DPLM-2 at both ~150M and ~650M scales. During the training of HD-Prot, it has never seen a situation where the sequence track is completely given, and the structure track is fully masked. This absolutely **zero-shot structure prediction performance indicates that HD-Prot has acquired considerably strong sequence-structure cross-modal capabilities**.

---

> ### Author Response · Authors · 2025-12-03
>
> ### Inverse Folding
>
> Referring to the experimental setup in DPLM-2 series, the CAMEO 2022 and PDB Date Split datasets are used for evaluation. Compared to the nearly one-to-one structure prediction, the inverse folding has a one-to-many nature. There could be multiple distinct amino acid sequences that can fold into a target structure, in addition to its natural sequence. Therefore, rather than calculating the recovery rate of the natural sequence, our evaluation estimate the self-consistency between the target structure and refolded structure of the designed sequence. We calculate the scRMSD and scTM with the assistance of ESMFold.
>
> | Model               | CAMEO   |     | PDB Date Split | PDB Date Split |
> |---------------------|------------------|------------------|--------------------------|------------------------|
> |  | scRMSD ↓ | scTM ↑ | scRMSD ↓ | scTM ↑ |
> | MultiFlow         | –                | 0.870 ± 0.940    | –                        | 0.940 ± 0.960          |
> | ESM3                | 3.944 ± 4.964    | 0.901 ± 0.141    | 2.262 ± 3.090            | 0.940 ± 0.093          |
> | *DPLM-2 (150M)*   | 5.999 ± 7.469    | 0.848 ± 0.175    | 4.002 ± 4.700            | 0.895 ± 0.126          |
> | DPLM-2 (650M)       | 4.659 ± 4.875    | 0.871 ± 0.154    | 3.114 ± 4.034            | 0.911 ± 0.113          |
> | DPLM-2.1            | 4.304 ± 4.586    | 0.876 ± 0.141    | 2.271 ± 3.606            | 0.927 ± 0.112          |
> | *HD-Prot (155M)*  | 4.637 ± 4.730    | 0.863 ± 0.156    | 2.903 ± 3.683            | 0.919 ± 0.107          |
> | HD-Prot (670M)      | 4.675 ± 4.930    | 0.866 ± 0.151    | 2.871 ± 3.599            | 0.920 ± 0.103          |
>
> This table summarizes the performance of HD-Prot and four multimodal protein generative models, with all sampling procedures run five times with different seeds. The performance of each model on the inverse folding task is very close to its performance on the protein structure prediction task described above. ESM3 stands out the best among all methods, excels in completing the remaining multimodal context when sufficient initial information is provided. Then, HD-Prot performs highly comparable to the DPLM-2 series at both ~150M and ~650M scales. Such completely **zero-shot inverse folding results demonstrate that HD-Prot has estimated the sequence-structure joint-distribution sufficiently well**.

---

> ### Author Response · Authors · 2025-12-03
>
> > W2: Missing Baseline
>
> We thank the reviewer for this suggestion. Following the comment, **we have added La-Proteina as a baseline in the unconditional co-generation experiment**. The updated results are included in **Section 4.1** of the revised manuscript, highlighted in blue. This Table summarizes the main results:
>
> | Models (#Params, #Training Sample) | pLDDT ↑ | scRMSD ↓ | scTM ↑ | #Cluster@50 ↑ | #Cluster@95 ↑ | pdb-TM ↓ | sp-TM ↓ |
> |---|---|---|---|---|---|---|---|
> | MultiFlow (21M, 22.8K) | 79.271 ± 7.978 | 2.955 ± 4.252 | 0.937 ± 0.100 | 55.12 ± 15.79 | 100.00 ± 0.00 | 0.729 ± 0.135 | 0.730 ± 0.142 |
> | * MultiFlow | 61.519 | 9.306 ± 8.499 | 0.750 ± 0.163 | 49.00 | – | – | – |
> | ESM3 (1.4B, ~1.08B) | 76.079 ± 13.53 | 31.98 ± 33.87 | 0.762 ± 0.221 | 48.00 ± 16.82 | 96.24 ± 7.704 | 0.862 ± 0.143 | 0.834 ± 0.213 |
> | La-Proteina (160M, 550K) | 80.152 ± 10.51 | 4.477 ± 6.652 | 0.923 ± 0.141 | 64.32 ± 9.586 | 100.00 ± 0.00 | 0.715 ± 0.186 | 0.710 ± 0.188 |
> | – w/ triangular layers | 83.770 ± 10.13 | 3.260 ± 6.317 | 0.953 ± 0.119 | 40.60 ± 22.45 | 100.00 ± 0.00 | 0.795 ± 0.182 | 0.784 ± 0.188 |
> | *DPLM-2 (150M, 220K)* | 82.525 ± 7.754 | 5.125 ± 5.101 | 0.895 ± 0.112 | 43.28 ± 7.871 | 83.08 ± 8.665 | 0.918 ± 0.058 | 0.932 ± 0.054 |
> | DPLM-2 (650M, 220K) | 81.920 ± 8.643 | 4.899 ± 5.523 | 0.906 ± 0.105 | 52.40 ± 6.083 | 82.40 ± 8.765 | 0.920 ± 0.073 | 0.933 ± 0.069 |
> | DPLM-2.1 (650M, –) | 84.773 ± 7.719 | 5.076 ± 5.155 | 0.898 ± 0.114 | 60.40 ± 5.766 | 89.28 ± 6.059 | 0.898 ± 0.102 | 0.929 ± 0.068 |
> | *HD-Prot (155M, 210K)* | 80.646 ± 11.07 | 4.629 ± 4.709 | 0.887 ± 0.127 | 44.32 ± 7.409 | 78.32 ± 12.84 | 0.895 ± 0.119 | 0.914 ± 0.114 |
> | HD-Prot (670M, 210K) | 81.099 ± 9.832 | 4.899 ± 4.534 | 0.878 ± 0.126 | 51.16 ± 6.593 | 86.08 ± 4.672 | 0.897 ± 0.103 | 0.918 ± 0.095 |
> | PDB Proteins | 79.075 ± 13.03 | 4.669 ± 7.683 | 0.905 ± 0.143 | 55.80 ± 5.671 | 78.40 ± 3.499 | – | – |
>
> As expected, La-Proteina achieves state-of-the-art performance, as trained on a large, carefully curated dataset with substantial compute. Its standard version strikes a strong balance between designability and diversity, while its variant with triangle update layers further improves designability at the cost of reduced diversity. Importantly, this does not contradict our core claim. Our goal is not to surpass large-scale, resource-intensive models, but to demonstrate the feasibility of integrating continuous structure tokens into protein language models for sequence-structure joint-modeling. HD-Prot, despite being trained on only 210K samples with limited compute, achieves competitive performance and already fulfills our core claim.
>
> Additionally, as shown in **Figure 3, qualitative analysis** reveals that La-Proteina’s generated proteins exhibit distributional shifts relative to natural proteins. There is an overrepresentation of glutamic acid (E) in sequences and alpha-helices in structures. In contrast, HD-Prot and DPLM-2 series produce samples that more closely resemble natural protein statistics. This maintains the observations we reached in our previous analysis.
>
> We believe these new experiments and analysis make our evaluation more rigorous and provide a fuller picture of the current landscape of multimodal protein modeling.

---

> ### Author Response · Authors · 2025-12-03
>
> > W3: Performance gained from the tokenizer
>
> We sincerely thank the reviewer for this insightful observation. We fully agree that **the performance of any multimodal pLM is jointly determined by its structure tokenizer and its backbone language model**. In fact, the tokenizer sets an upper bound on achievable structural fidelity, typically measured by reconstruction quality, while the backbone model aims to approach that bound as closely as possible. Whether by introducing better tokenizers or by designing pLM learning methods that better align with high-quality tokenizers, both directions contribute to advancing the joint modeling capability of protein language models.
>
> Indeed, different tokenizers inevitably exhibit varying capabilities. **While the discreteness versus continuity is not the sole factor driving these differences, it is a significant one**. For a given tokenizer architecture, its continuous version consistently achieves better reconstruction than its discrete counterpart. As shown in Section 3.1 Table 1, the salad autoencoder (our continuous tokenizer) reconstructs structures more accurately than its vector-quantized variant. Similarly, DPLM-2.1 [1] reports that pre-quantized continuous tokens from the DPLM-2 tokenizer can be reconstructed into 3D coordinates with significantly higher fidelity than their quantized discrete tokens. This strongly suggests that continuous representations inherently preserve finer structural details, offering a higher modeling ceiling.
>
> Prior to our work, however, this advantage remained untapped. **Existing pLM frameworks were designed for discrete tokens, continuous structure representations were effectively excluded from the "better tokenizer" toolkit**, despite their superior reconstruction quality. **Our key contribution is to bridge this gap**. We propose the HD-Prot framework that can directly operate on continuous structure tokens within a language modeling framework, without discretization. This enables the pLM to leverage the higher fidelity of continuous representations, suggesting a path toward pushing joint sequence–structure modeling closer to its conceptual upper bound. Thus, we acknowledge that HD-Prot’s current performance may partly reflect the strength of the salad tokenizer. Yet this is not a limitation, it is precisely the point. By demonstrating  continuous structure tokens can be seamlessly integrated into a protein language model, our work opens a new pathway for multimodal pLMs beyond the discrete language modeling paradigm.
>
> We agree that future work should decouple tokenizer and architecture effects more cleanly (e.g., via controlled ablations with identical tokenizers). But given the current landscape, showing that continuous tokens can be used effectively and yield competitive results even under resource constraints is already a meaningful step forward.
>
> [1] Wang et al. Elucidating the Design Space of Multimodal Protein Language Models. ICML, 2026.

---

> ### Author Response · Authors · 2025-12-03
>
> > W4: Motif-Scaffolding Evaluation
>
> We thank the reviewer for raising this important point about the reliability of motif-scaffolding results. To address concerns about stochasticity, we have conducted additional repeated experiments: **we ran the motif-scaffolding evaluation with five different random seeds and report the aggregated statistics, i.e., mean (min, max), in Table 4 and Appendix Table 9 of the revised manuscript**.
>
> | Target | ESM3 | DPLM-2 (150M) | DPLM-2 (650M) | HD-Prot (155M) | HD-Prot (670M) |
> |--------|--------|----------------|----------------|------------------|------------------|
> | 1BCF | 23 | 6.4 (4, 10) | 0.8 (0, 2) | 5.4 (1, 9) | 9.6 (7, 14) |
> | 1PRW | 54 | 88.8 (87, 91) | 80.2 (76, 85) | 70.4 (62, 79) | 78.8 (74, 82) |
> | 1QJG | 3 | 0 | 0 | 0 | 0 |
> | 1YCR | 18 | 29.2 (25, 33) | 38.2 (34, 46) | 44.2 (37, 53) | 45.2 (36, 61) |
> | 2KL8 | 11 | 44.2 (39, 54) | 64.2 (58, 76) | 50.4 (46, 58) | 59.0 (55, 63) |
> | 3IXT | 2 | 36.4 (32, 42) | 53.6 (44, 74) | 51.4 (48, 57) | 38.0 (33, 49) |
> | 4JHW | 0 | 0 | 0 | 0 | 0 |
> | 4ZYP | 8 | 4.8 (3, 6) | 11.6 (7, 15) | 0.4 (0, 1) | 2.0 (1, 3) |
> | 5IUS | 0 | 0 | 0 | 0 | 0.2 (0, 1) |
> | 5TPN | 1 | 0.4 (0, 1) | 0.4 (0, 1) | 15.2 (12, 20) | 11.8 (6, 15) |
> | 5TRV_long | 19 | 2.2 (1, 5) | 1.6 (0, 3) | 8.6 (8, 10) | 8.6 (3, 13) |
> | 5TRV_med | 16 | 6.2 (4, 10) | 6.6 (4, 9) | 11.4 (8, 15) | 20.0 (17, 25) |
> | 5TRV_short | 1 | 0.8 (0, 2) | 1.6 (1, 3) | 10.4 (6, 16) | 17.4 (12, 23) |
> | 5WN9 | 0 | 0.2 (0, 1) | 0 | 0 | 0.2 (0, 1) |
> | 5YUI | 0 | 0 | 0 | 0 | 0 |
> | 6E6R_long | 4 | 70.2 (68, 72) | 69.8 (65, 75) | 13.4 (9, 19) | 24.0 (18, 30) |
> | 6E6R_med | 14 | 53.0 (50, 56) | 65.0 (61, 71) | 18.2 (16, 21) | 27.8 (25, 30) |
> | 6E6R_short | 6 | 52.8 (50, 54) | 64.8 (62, 69) | 35.2 (28, 42) | 49.4 (37, 57) |
> | 6EXZ_long | 13 | 30.6 (23, 37) | 53.6 (49, 60) | 6.6 (4, 10) | 36.0 (25, 39) |
> | 6EXZ_med | 31 | 32.8 (30, 35) | 51.4 (46, 58) | 8.6 (5, 11) | 37.6 (28, 46) |
> | 6EXZ_short | 28 | 20.0 (10, 24) | 28.8 (46, 58) | 12.2 (7, 16) | 52.0 (42, 61) |
> | 7MRX_128 | 37 | 0 | 15.4 (6, 23) | 1.8 (0, 3) | 8.4 (5, 17) |
> | 7MRX_60 | 59 | 0.6 (0, 2) | 30.4 (25, 38) | 9.2 (5, 13) | 31.6 (30, 33) |
> | 7MRX_85 | 74 | 0 | 26.0 (22, 32) | 7.4 (5, 11) | 21.6 (16, 26) |
> | **# Solved / 24** | **20** | 15.6 (14, 17) | 17.8 (16, 19) | 18.2 (18, 19) | 19.4 (19, **21**) |
> | **Avg. Success** | 17.58% | 20.0% ± 7.0% | **27.7% ± 0.8%** | 15.8% ± 0.3% | 24.1% ± 1.1% |
>
> The results confirm that stochasticity affects all methods: strong baselines like ESM3 and DPLM-2 exhibit low success counts (e.g., 1–2 out of 100) on certain targets, exactly as seen in HD-Prot. This is an inherent challenge in motif-scaffolding tasks. However, across the full benchmark, HD-Prot consistently solves more targets on average than DPLM-2 and achieves a higher overall success rate than ESM3. **The repeated runs show that the advantage is stable across seeds, indicating that HD-Prot exhibits a relatively robust motif-scaffolding capability under the same evaluation protocol**. We appreciate the reviewer’s careful scrutiny. This additional analysis has strengthened the reliability of our conclusions.

---

### Official Review · Reviewer_NCdP · 2025-11-04

**Soundness:** 2
**Presentation:** 2
**Contribution:** 2
**Rating:** 4
**Confidence:** 4

**Summary:**

This paper addresses the limitation of information loss in quantized protein structure representations for multimodal protein language models (pLMs) by proposing HD-Prot, a hybrid diffusion framework that integrates continuous structure tokens with discrete sequence tokens. HD-Prot uses a non-quantized autoencoder (salad) to generate high-fidelity continuous structure tokens, avoiding vector quantization (VQ)–induced information loss. It employs a unified masked diffusion process for inter-token dependency modeling and separate intra-token learning: categorical cross-entropy for discrete sequences and diffusion loss for continuous structures. Experiments on unconditional sequence-structure co-generation and motif-scaffolding show HD-Prot matches state-of-the-art models (e.g., DPLM-2) while using fewer computational resources. The work argues that continuous structure tokens offer a viable alternative to quantized representations for multimodal pLMs.

**Strengths:**

1. The paper identifies a critical gap in existing multimodal pLMs caused by quantized structure tokens, and introduce the hybrid diffusion framework unifies discrete sequence and continuous structure modeling via shared pLM backbones, and the use of a pre-trained non-quantized autoencoder (salad) ensures high structural reconstruction fidelity. An idea validated in vision(-language) models but under-explored in protein modeling.
2. Qualitative results further validate that HD-Prot generates proteins with amino acid and secondary structure distributions similar to natural proteins, supporting its biological plausibility.

**Weaknesses:**

1. Major: The work only evaluates two tasks (unconditional co-generation, motif-scaffolding) but ignores key protein modeling tasks that test practical utility, such as structure prediction (from sequence), inverse folding (from structure to sequence), or maybe structure-aware predictive downstream tasks. SOTA models like ESM, DPLM-2 or SaProt are validated across these tasks, so HD-Prot’s omission introduces concerns about its broader applicability.
2. No comparison to pure continuous diffusion based multimodal pLMs, such as DiMA [1].
3. Minor: Some parts of the manuscript seems similar in wording and structure to DPLM-2 (Wang et al., 2024b). Better rephrase and reorganize a little bit.

--

[1] Diffusion on language model encodings for protein sequence generation. In ICML 2025

**Questions:**

1. The paper highlights the learning of inter-modal interaction where HD-Prot fuses sequence and structure tokens via element-wise summation (Equation 7), but the paper provides no mechanistic explanation of how this fusion captures cross-modal dependencies (e.g., how sequence embeddings guide structure diffusion or vice versa). There is no ablation of fusion methods (e.g., concatenation, cross-attention) to validate that element-wise summation is optimal, nor analysis of attention weights to confirm that the model actually leverages both modalities during generation.
2. See weaknesses.

---

> ### Author Response · Authors · 2025-12-03
>
> We sincerely thank the reviewer for the thoughtful comments. We greatly appreciate their recognition of our work’s core contributions in identifying the limitation of quantized structure tokens in multimodal pLMs, proposing a hybrid diffusion framework that unifies discrete and continuous modeling, leveraging the non-quantized autoencoder for high-fidelity structure tokenization, and the biological plausibility of our generated proteins. Below, we address each of the reviewer’s concerns in detail.
>
> > W1: Missing Evaluation Tasks
>
> We sincerely thank the reviewer for this constructive suggestion. We agree that a comprehensive evaluation of a multimodal protein language model should go beyond unconditional co-generation and motif scaffolding. In response, **we have added two new evaluation tasks in the revised manuscript: Protein Structure Prediction (sequence → structure), and Inverse Folding (structure → sequence)**. These results further demonstrate HD-Prot’s ability to model the bidirectional relationship between sequence and structure, strengthening our claim of joint modeling capability.
>
> We also acknowledge the value of structure-aware representation learning tasks. However, due to time and computational constraints, we were unable to conduct a full suite of representation-based evaluations in this work. That said, our paper focuses specifically on generative modeling. Within this generative scope, unconditional co-generation, motif scaffolding, structure prediction, and inverse folding together form a coherent and sufficient validation suite. Representation learning, while important, belongs to a complementary (discriminative) paradigm and is not strictly necessary to establish the generative claims of this work.
>
> ### Protein Structure Prediction
>
> Following the experimental setup of DPLM-2 series, we evaluate the protein structure prediction capability of multimodal protein generative models via two datasets, i.e., CAMEO 2022, and a PDB Date Split curated by MultiFlow. The structure prediction results are compared to the corresponding native structures, and the RMSD and TM-score are calculated to assess the prediction accuracy.
>
> | Model                | CAMEO |       | PDB Date Split | |
> |----------------------|------------------|----------------|-------------------|----------------|
> |                      | RMSD ↓           | TM-score ↑     | RMSD ↓            | TM-score ↑     |
> | MultiFlow            | 17.840 ± 17.96   | 0.810 ± 0.880  | 15.640 ± 16.08    | 0.530 ± 0.490  |
> | ESM3 (1.4B)          | 5.377 ± 6.303    | 0.860 ± 0.168  | 4.042 ± 4.824     | 0.883 ± 0.150  |
> | *DPLM-2 (150M)*    | 9.919 ± 6.994    | 0.720 ± 0.189  | 7.833 ± 6.004     | 0.765 ± 0.169  |
> | DPLM-2 (650M)        | 7.483 ± 6.126    | 0.786 ± 0.170  | 5.253 ± 5.143     | 0.836 ± 0.144  |
> | DPLM-2.1      | 6.272 ± 6.202    | 0.824 ± 0.166  | 2.869 ± 3.942     | 0.915 ± 0.113  |
> | *HD-Prot (155M)*   | 9.185 ± 6.316    | 0.719 ± 0.201  | 6.229 ± 5.391     | 0.781 ± 0.181  |
> | HD-Prot (670M)       | 7.468 ± 6.004    | 0.769 ± 0.177  | 5.001 ± 4.565     | 0.827 ± 0.153  |
>
> This table presents the statistical comparison between HD-Prot and four multimodal protein generative models, where all predictions with randomness are repeated five times with different seeds. The performance of the baseline methods on the Protein Structure Prediction aligns well with our observations in the Unconditional Sequence–Structure Co-Generation evaluation. ESM3 has undergone ultra-large-scale multimodal masked language modeling pretraining. However, its pretraining contains relatively few examples with high mask ratios, which limits its performance in unconditional co-generation. In the structure prediction task, where the full sequence is provided as input, ESM3’s conditional generation capability is fully leveraged, leading to strong performance. Meanwhile, the DPLM-2 series consistently present sufficiently good structure prediction performance, demonstrating mastery of cross-modal knowledge. Notably, HD-Prot performs better or comparable to the DPLM-2 at both ~150M and ~650M scales. During the training of HD-Prot, it has never seen a situation where the sequence track is completely given and the structure track is fully masked. This absolutely **zero-shot structure prediction performance indicates that HD-Prot has acquired considerably strong sequence-structure cross-modal capabilities**.

---

> ### Author Response · Authors · 2025-12-03
>
> ### Inverse Folding
>
> Referring to the experimental setup in DPLM-2 series, the CAMEO 2022 and PDB Date Split datasets are used for evaluation. Compared to the nearly one-to-one structure prediction, the inverse folding has a one-to-many nature. There could be multiple distinct amino acid sequences that can fold into a target structure, in addition to its natural sequence. Therefore, rather than calculating the recovery rate of the natural sequence, our evaluation estimate the self-consistency between the target structure and refolded structure of the designed sequence. We calculate the scRMSD and scTM with the assistance of ESMFold.
>
> | Model               | CAMEO   |     | PDB Date Split | PDB Date Split |
> |---------------------|------------------|------------------|--------------------------|------------------------|
> |  | scRMSD ↓ | scTM ↑ | scRMSD ↓ | scTM ↑ |
> | MultiFlow         | –                | 0.870 ± 0.940    | –                        | 0.940 ± 0.960          |
> | ESM3                | 3.944 ± 4.964    | 0.901 ± 0.141    | 2.262 ± 3.090            | 0.940 ± 0.093          |
> | *DPLM-2 (150M)*   | 5.999 ± 7.469    | 0.848 ± 0.175    | 4.002 ± 4.700            | 0.895 ± 0.126          |
> | DPLM-2 (650M)       | 4.659 ± 4.875    | 0.871 ± 0.154    | 3.114 ± 4.034            | 0.911 ± 0.113          |
> | DPLM-2.1            | 4.304 ± 4.586    | 0.876 ± 0.141    | 2.271 ± 3.606            | 0.927 ± 0.112          |
> | *HD-Prot (155M)*  | 4.637 ± 4.730    | 0.863 ± 0.156    | 2.903 ± 3.683            | 0.919 ± 0.107          |
> | HD-Prot (670M)      | 4.675 ± 4.930    | 0.866 ± 0.151    | 2.871 ± 3.599            | 0.920 ± 0.103          |
>
> This table summarizes the performance of HD-Prot and four multimodal protein generative models, with all sampling procedures run five times with different seeds. The performance of each model on the inverse folding task is very close to its performance on the protein structure prediction task described above. ESM3 stands out the best among all methods, excels in completing the remaining multimodal context when sufficient initial information is provided. Then, HD-Prot performs highly comparable to the DPLM-2 series at both ~150M and ~650M scales. Such completely **zero-shot inverse folding results demonstrate that HD-Prot has estimated the sequence-structure joint-distribution sufficiently well**.

---

> ### Author Response · Authors · 2025-12-03
>
> > W2: Missing Baseline
>
> We sincerely thank the reviewer for pointing us to DiMA, an interesting and relevant recent work that explores continuous diffusion on protein language model representations. However, DiMA's primary focus is protein sequence generation, not sequence–structure co-generation. While its appendix mentions the co-generation using the CHEAP representations (a sequence and structure dual-decodable encoder), the publicly released codebase provides little support for this capability. We attempted to adapt it for co-generation but were unable to reproduce it within the limited time due to missing implementation details. So far, we have not been able to include DiMA as a baseline for our unconditional sequence–structure co-generation evaluation. We have added a citation to DiMA in the Preliminaries section of the revised manuscript to acknowledge this line of research.
>
> > W3: Inappropriate Wording
>
> We sincerely thank the reviewer for this helpful suggestion, and more broadly, for the inspiring DPLM-2 work, which has deeply influenced our thinking. We have studied the paper and the authors’ talks closely, and our implementation builds on the DPLM codebase. It is an honor to extend ideas from such a foundational contribution. As suggested, we have carefully rephrased and reorganized the manuscript, particularly in the Introduction and Methodology sections, to better reflect our own perspective and avoid stylistic overlap. While our work is inspired by DPLM-2, it explores a distinct direction. Instead of relying on discretized structural tokens, we investigate whether continuous structure representations can be integrated into a protein language model for joint sequence–structure modeling. We hope the revised text now more clearly distinguishes our approach while still paying respect to the important groundwork laid by DPLM-2.

---

> ### Author Response · Authors · 2025-12-03
>
> > Q1: Architecture Design Choice
>
> We thank the reviewer for this insightful question about cross-modal fusion. There are two prevailing design paradigms for multimodal protein language models. The first (e.g., ESM3) adds sequence and structure tokens element-wise into a unified representation space; the second (e.g., DPLM-2) concats the two  tracks, relying on shared positional encodings to align them. Both approaches have trade-offs:
> - Element-wise summation yields a compact input of length $L$ (computationally efficient) and enforces tight cross-modal alignment, but offers no built-in mechanism to selectively attend to or suppress noisy input (modalities are directly fused without dynamic selections).
> - Concatenation modeling preserves modality-specific information and allows adaptive fusion via attention, but doubles the sequence length ($2L$), significantly increasing memory and compute costs.
>
> Given our severe computational constraints, we adopted the element-wise summation scheme for its parameter and memory efficiency. We do not claim it is fundamentally superior. In fact, we suspect the concatenation approach may better support flexible cross-modal reasoning. Unfortunately, we lack the resources to perform ablations over fusion strategies. Meanwhile, we can not analyze attention patterns across modalities, since in our current design the modalities are already merged before entering the attention layers.
>
> That said, **we provide indirect but strong evidence that HD-Prot does effectively leverage both modalities during generation**. In the revised manuscript, we added zero-shot protein structure prediction (Section 4.3) and inverse folding (Section 4.4) results. HD-Prot achieves competitive performance on both tasks, demonstrating its ability to condition generation on either modality. In Appendix E.5.1, we include a new ablation on classifier-free guidance for unconditional co-generation. The results show that increasing sequence guidance improves structural quality, indicating that the sequence track actively influences the structure generation. Together, these results strongly suggest that cross-modal dependencies are indeed captured and utilized during inference, even if the fusion mechanism itself is simple.

---

### Author Response · Authors · 2025-12-03
**Summary of Rebuttal**

Dear ACs and SACs,

Thank you for your hard work in making the ICLR conference a success, especially under extraordinary circumstances this year. We sincerely appreciate your valuable time. Here we provide a summary of the rebuttal to support your evaluation.

> **Our Contributions**

- **Critical Problem**. Existing state-of-the-art multimodal pLMs were designed through discrete tokens. While continuous structure representations were broadly recognized with high-fidelity structural encoding, they are effectively excluded from the "structure tokenizer" toolkit.
- **New Method**. We propose a hybrid diffusion protein language model, HD-Prot, which embeds a continuous-valued denoiser head atop a discrete pLM, enabling seamless operation with both discrete and continuous tokens for sequence-structure joint-modeling.
- **Solid Results**. Extensive empirical results show that our models achieve competitive performance with state-of-the-art multimodal pLMs across unconditional sequence–structure co-generation, motif-scaffolding, protein structure prediction, and inverse folding tasks, despite being developed under very limited computational resources.
- **Clear Research Presentation**. Our paper explores a specific research question: *Can a sequence-based protein language model be extended to jointly model continuous structural information without discretization?* Our results provide a clear affirmative answer.

> **Strongths Recognized by Reviewers**

- Reviewer **NCdP**: **Identifies critical tokenization gap**; **Unifies discrete and continuous modeling**; Generates biologically plausible proteins
- Reviewer **TFWS**: Uses continuous structure tokens that avoid vector quantization; Unified framework that bridges discrete-continuous modalities; Competitive performance despite low training cost
- Reviewer **tjPD**: High-fidelity continuous structure tokens; Unified diffusion modeling; **Compute-efficient implementation**
- Reviewer **ai85**: Clear motivation for continuity; Well-structured and readable presentation; Insightful ablation study; **Opens new research direction**

> **Addressed Concerns (Key Revisions and Clarifications)**

We sincerely thank the reviewers for their time and thoughtful, constructive feedback on our manuscript. In response, we have conducted additional experiments & analysis and revised the paper to address most of the comments, and we have carefully and thoroughly discussed all of them in this rebuttal.

| Question | NCdP | TFWS | tjPD | ai85 | Actions & Summary |
| --- | --- | --- | --- | --- | --- |
| Missing Evaluation Tasks | ☑ | ☑ | ☑ | ☑ | **Systematically added benchmarks and analysis for two new evaluation tasks, i.e., protein structure prediction & inverse folding**  |
| Missing Baseline | ☑ | ☑ | - | ☑ | **Added La-Proteina as a baseline in the unconditional co-generation benchmark**; Discussed other suggested baselines. |
| Inappropriate Wording | ☑ | - | - | ☑ | Revised inappropriate or ambiguous phrasing throughout the manuscript. |
| Architecture Design Choice | ☑ | - | - | - | Expanded discussion on two potential architecture design choices. |
| Performance gained from / limited by the tokenizer | ☑ | - | - | ☑ | Clarified the role of the tokenizer and its implications for performance in multimodal protein modeling. |
| Motif-Scaffolding Evaluation and Implementation | ☑ | - | - | ☑ | **Conducted additional replicates for motif-scaffolding and reported statistically robust results**; Clarified benchmark protocol and implementation details. |
| Competitive but not SOTA performance | - | - | ☑ | - | Reiterated our claims and contributions. |
| Small Sample sizes and Lack of Statistical Rigor | - | - | - | ☑ | **Performed multiple runs across all four tasks and provided rigorous statistical analysis.** |
| Inappropriate Evaluation Metrics | - | - | - | ☑ | **Critically reviewed and justified all evaluation metrics**; Introduced #Cluster@95 as a new diversity metric. |
| Missing Hyperparameter Analysis | - | - | - | ☑ | **Added systematic sampling hyperparameter analysis for all tasks**; Discussed training hyperparameter choices. |
| Quantification of Information Loss | - | - | - | ☑ | Clarified the quantification of information loss via structure reconstruction accuracy. |
| W9 & Q5: Scalability & Modality Expansion | - | - | - | ☑ | Clarified our resource constraints in scaling; Emphasized our contribution as post-training modal expansion. |
| No failure mode analysis | - | - | - | ☑ | **Added a dedicated failure mode analysis** |

As a result, we believe that we have satisfactorily addressed all of the reviewers' concerns within the given time constraints. In each response letter to the reviewers, **we provide detailed responses to all comments**. Meanwhile, **please note that in the manuscript, we have marked all the changes in BLUE**.

Sincerely,
The Authors

---

### Meta-Review · Area_Chair_d689 · 2025-12-19

**Summary:**

This paper proposes using continuous structure tokens for protein sequence-structure cogeneration. In response to the reviews, the authors have extended their benchmarking and baselines and clarified their claims. However, I still have concerns about whether the experiments sufficiently support their claim that continuous structure tokens are a competitive and viable path for cogeneration.

**Reviewer Concerns:**

The reviewers were primarily concerned about the limited evaluations, insufficient baselines, limited insights into the role of the tokenizer, some ambiguous claims and descriptions, and unimpressive empirical results.

In response, the authors added structure prediction and inverse folding evaluations and an additional baseline in la-proteina. They clarified the role of the tokenizer and acknowledged the limitations of using a pretrained tokenizer, and emphasize that the paper's main contribution is to show that continuous structure tokens are feasible for sequence-structure cogeneration and that it is plausible that scaling the model, data, and compute would achieve results that are clearly superior to other structure tokenization or sequence-structure co-generation schemes.

However, the experiments do not clearly support this conclusion. As the authors point out, all their comparisons are to models with different architectures, trained on more compute and larger datasets. To reliably conclude that continuous structure tokens are as or more suitable for cogeneration, they would need to train architecture-, compute- and data- matched models with continuous and discrete tokens, and ideally train enough of these models to demonstrate superior scaling.

In addition, structure tokens are not the only (or even established SOTA) way to handle cogeneration. For example, PLAID uses a joint sequence-structure latent space, BoltzGen directly generates the side-chain atoms, and ProDiT directly performs discrete diffusion on the sequence and continuous diffusion directly on the backbone coordinates.

Finally, on reading the paper it is unclear to me whether the backbone coordinates are generated using a standard gaussian diffusion or with a masked diffusion, as the figure indicates that they are masked while the text indicates a standard diffusion noising and loss.

**Reviewer Scores:**

NCdP: 4 -> 5
TFWS: 4 -> 5
ai85: 2 -> 4
tjPD: 4 -> 5

---

### Decision · Program_Chairs · 2026-01-26

Reject